# Neuron–non-neuron electrical coupling networks are involved in chronic stress-induced electrophysiological changes in lateral habenular neurons

Kenji Yamaoka[1] (ID), Kanako Nozaki[2], Meina Zhu[2], Haruhi Terai[2], Kenta Kobayashi[3], Hikaru Ito[2], Miho Matsumata[2], Hidenori Takemoto[2], Shinya Ikeda[4], Yusuke Sotomaru[4], Tetsuji Mori[5], Hidenori Aizawa[2] and Kouichi Hashimoto[1] (ID)

[1] *Department of Neurophysiology, Graduate School of Biomedical and Health Sciences, Hiroshima University, Hiroshima, Japan*
[2] *Department of Neurobiology, Graduate School of Biomedical and Health Sciences, Hiroshima University, Hiroshima, Japan*
[3] *Section of Viral Vector Development, National Institute for Physiological Sciences, Okazaki, Japan*
[4] *Natural Science Center for Basic Research and Development, Hiroshima University, Hiroshima, Japan*
[5] *Department of Biological Regulation, School of Health Science, Faculty of Medicine, Tottori University, Yonago, Japan*

Handling Editors: Katalin Toth & I-Shan Chen

The peer review history is available in the Supporting Information section of this article (https://doi.org/10.1113/JP287286#support-information-section).

**The Journal of Physiology**

**Abstract figure legend** We investigated electrophysiological changes in lateral habenula (LHb) neurons induced by chronic social defeat stress in mice. LHb neurons exhibited short (<400 ms) rebound depolarizing potentials (short-RDPs) or long-RDPs (order of seconds) (long-RDPs) after the offset of hyperpolarization. The incidence of long-RDP neurons was significantly reduced in the stress susceptible mice. The long depolarizing phase of long-RDPs was mediated by cyclic nucleotide-gated (CNG) channels. The expression of *Cnga4*, a subtype of the CNG channel, was decreased in the stress-susceptible mice. Our analyses revealed that LHb neurons formed functional networks with non-neuronal cells, and CNG channels were activated in the neuron–non-neuron networks via gap junctions.

**Abstract** The lateral habenula (LHb) is a key brain structure that receives input from higher brain regions and regulates monoaminergic activity. LHb hyperactivity has been implicated in the pathophysiology of depression, but the electrophysiological mechanisms underlying this hyperactivity remain poorly understood. To address this issue, we investigated how chronic stress alters the firing properties of LHb neurons in a mouse model of chronic social defeat. Whole-cell recordings were conducted from LHb neurons in the mouse acute brain slices. LHb neurons exhibited two types of rebound depolarizing potentials (RDPs) after the offset of hyperpolarization: short-RDPs (lasting <400 ms) and long-RDPs (order of seconds). Stress-susceptible mice showed a significantly reduced occurrence of long-RDPs, whereas spike firing in response to depolarizing current injections remained unchanged. Both short- and long-RDPs were triggered by T-type voltage-dependent $Ca^{2+}$ channels and shortened by small-conductance $Ca^{2+}$-activated $K^+$ (SK) channels. The prolonged depolarizing phase of long-RDPs was mediated by cyclic nucleotide-gated (CNG) channels, which were activated via electrical coupling formed between neurons and non-neuronal cells. Whole-cell recording using an internal solution including a gap junction-permeable dye revealed that neurons formed dye coupling with non-neuronal cells, including oligodendrocytes and/or oligodendrocyte precursor cells. RNA-sequencing and genome editing experiments suggested that *Cnga4*, a CNG channel subtype, was the primary candidate for the long depolarizing phase of long-RDP, and its expression was decreased in the stress-susceptible mice. These findings suggest that stress-dependent changes in the firing activity of neurons are regulated by neuron–non-neuron networks formed in the LHb.

(Received 11 July 2024; accepted after revision 20 February 2025; first published online 1 April 2025)

**Corresponding author** K. Hashimoto: Department of Neurophysiology, Graduate School of Biomedical and Health Sciences, Hiroshima University; Hiroshima 734-8553, Japan. Email: hashik@hiroshima-u.ac.jp

## Key points

- Mouse lateral habenular (LHb) neurons exhibit short (<400 ms) rebound depolarizing potentials (short-RDPs) or long-RDPs (order of seconds) (long-RDPs) after the offset of hyperpolarization.
- The incidence of long-RDP neurons is significantly reduced in mice susceptible to chronic social defeat stress.
- The long depolarizing phase of long-RDPs is mediated by cyclic nucleotide-gated (CNG) channels, which are activated in non-neuronal cells via gap junctions.
- The expression of *Cnga4*, the gene encoding a subtype of the CNG channel, is decreased in the stress-susceptible mice.
- These results help us understand the mechanisms underlying stress-induced electrophysiological changes in LHb neurons and the functional roles of neuron–non-neuron networks for these neurons.

**Kenji Yamaoka** has been actively involved in research since his time at Hiroshima University School of Medicine. After graduating, he worked as a psychiatrist for several years before transitioning to a career in research. Currently, he focuses on advancing the understanding of the pathophysiology of psychiatric disorders as a graduate student with a particular interest in the roles of the lateral habenula. His research aims to explore the complex interactions between neurons and glia in psychiatric conditions, contributing to a better understanding of their underlying mechanisms.

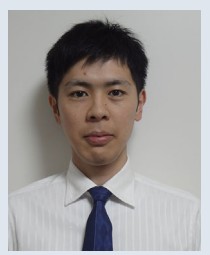

## Introduction

The lateral habenular (LHb) has been implicated in the pathophysiology of psychiatric disorders (Aizawa & Zhu, 2019; Ellison, 1994; Sandyk, 1991; Sutherland, 1982). Glutamatergic LHb neurons receive input from the forebrain, hypothalamus and basal ganglia, and project to monoaminergic nuclei (Herkenham & Nauta, 1977, 1979; Wang & Aghajanian, 1977). Because the LHb provides indirect inhibitory projections to monoaminergic neurons (Jhou et al., 2009; Metzger et al., 2017; Yang et al., 2008), hyperactivation of LHb neurons is considered to suppress monoaminergic activity. Accumulating evidence has supported a close relationship between the hyper-activation of LHb neurons and depression-like behaviour in humans and animal models (Lecca et al., 2016; Li et al., 2013; Yang et al., 2018). Cui et al. (2018) demonstrated that the stress-induced firing changes are caused by indirect neuron–astrocyte interactions in rodents. Their findings suggest that upregulation of Kir4.1 in astrocytes reduces the local extracellular $K^+$ concentration around the somata of neurons, which hyperpolarizes the resting membrane potential of LHb neurons. This hyperpolarization promotes spontaneous burst firing in LHb neurons, a key feature associated with stress-induced hyperactivation (Cui et al., 2018; Yang et al., 2018). Despite these advances, the precise alterations in the intrinsic membrane excitability of LHb neurons under stress remain poorly understood. Because spike firing is fundamentally controlled by the activity of voltage-gated ion channels expressed in neurons, stress-induced changes in such ion channel expression or function could underlie firing pattern alterations. Changes in intrinsic ion channel properties in neurons could also be a basis of stress-induced hyperactivation of LHb neurons.

In the brains of mood disorder patients, a reduced density of glial cells has been observed (Cotter et al., 2002; Miguel-Hidalgo et al., 2010; Si et al., 2004; Uranova et al., 2004), along with diminished expression of various types of glial connexins (e.g. Cx43 and Cx30 in astrocytes, and Cx32 and Cx47 in oligodendrocytes) (Miguel-Hidalgo, 2022; Nagy et al., 2017; Okada et al., 2020; Ren et al., 2018; Tanti et al., 2019). Animal models of depression also exhibit reduced connexin expression (Huang et al., 2019; Miguel-Hidalgo, 2022; Miguel-Hidalgo et al., 2018; Okada et al., 2020), which is partially restored following anti-depressant treatment (Sun et al., 2012). These data suggest that disruptions in glial gap junction networks in the brain contribute to depressive behaviours. A decrease in various types of connexins suggests the disruption of intricate networks formed by multiple types of glial cells. Astrocytes and oligodendrocytes exhibit coupling mediated by gap junctions among not only homologous, but also heterologous partners (pan-glial syncytium) (Basu & Sarma, 2018; Miguel-Hidalgo, 2023; Nagy & Rash, 2000; Orthmann-Murphy et al., 2008; Peng et al., 2022). Moreover, these disturbances might not be confined to only glial cells. Electrical coupling between neurons and glial cells in certain brain regions has also been reported (Alvarez-Maubecin et al., 2000; Pakhotin & Verkhratsky, 2005). However, the details of the formation, regulation and functional significance of the networks involved in stress-induced responses remain poorly understood.

In the present study, we examined how social defeat stress affects the excitable membrane properties of LHb neurons using whole-cell recordings from LHb neurons in acute mouse brain slices. We found that LHb neurons exhibited diverse spike-firing patterns in response to depolarizing and hyperpolarizing current injections. LHb neurons displayed short rebound depolarizing potentials (short-RDPs) or long-RDPs that sometimes persisted for several seconds (long-RDPs) in response to hyperpolarizing current injections. Social defeat stress significantly decreased the proportion of neurons with long-RDPs in susceptible mice, whereas no such change was observed in resilient mice. Pharmacological and RNA-sequencing experiments suggested that this change was mediated by the downregulation of cyclic nucleotide-gated (CNG) channels involving the *Cnga4* subunit. Additionally, LHb neurons were found to form dye-coupling networks with neuronal and non-neuronal cells, which participate in the generation of the prolonged depolarization phase of long-RDPs. These data suggest that the spike firing of LHb neurons is regulated by the neuron–non-neuron networks formed within the LHb in a stress-dependent manner.

## Methods

### Ethical approval

All the animal experiments were authorized by the experimental animal ethics committee (A24-155) and the biosafety committee for living-modified organisms (#2024-229) of Hiroshima University. Male C57BL/6J aged 7–12 weeks and male retired-breeder ICR mice were obtained from CREA Japan (Tokyo, Japan) and Japan SLC (Hamamatsu, Japan), respectively. All the mice were maintained under specific-pathogen-free conditions under a 12:12 h light/dark photocycle (lights off at 20.00 h) with free access to food and water.

The mice were chosen from individual groups without special selection. For electrophysiology, coronal brain slices were prepared from C57BL/6J mice. The mice were placed in a chamber and the $CO_2$ level was subsequently increased. After the mice lost consciousness,

they were decapitated. For double whole-cell experiments, the mice were anaesthetized by an I.P. injection of a mixture of medetomidine hydrochloride (1.5 mg kg$^{-1}$; Domitor; Nippon Zenyaku Kogyo, Fukushima, Japan), midazolam (8 mg kg$^{-1}$; Sandoz; Sandoz, Tokyo, Japan) and butorphanol (10 mg kg$^{-1}$; Vetorphale; Meiji Seika Pharma, Tokyo, Japan) dissolved in saline. Following achievement of complete anaesthesia and analgesia, the mice were perfused transcardially with an ice-cold cutting solution containing (in mM): 215 sucrose, 2.5 KCl, 26 NaHCO$_3$, 1.25 NaH$_2$PO$_4$, 20 glucose, 5 sodium pyrvate, 1 CaCl$_2$ and 7 MgCl$_2$ (Velez-Fort et al., 2010) or *N*-methyl-D-glucamine (NMDG) solution containing (in mM): 92 NMDG, 2.5 KCl, 1.2 NaH$_2$PO$_4$, 30 NaHCO$_3$, 20 HEPES, 25 glucose, 5 sodium ascorbate, 2 thiourea, 3 sodium pyruvate, 10 MgSO$_4$ and 0.5 CaCl$_2$, titrated to pH 7.3–7.4 using concentrated HCl (Ting et al., 2018). A block of the brain was embedded in 2% (w/v) agarose and 300 μm coronal slices were prepared. For immuno-chemistry, C57BL/6J mice were deeply anaesthetized using the same anaesthetic mixture used for the double whole-cell experiments. Mouse tissues were fixed by trans-cardial perfusion using either 9% glyoxal and 8% acetic acid (pH 4, adjusted with NaOH) after pre-perfusion with 0.9% saline (Konno et al., 2023; Richter et al., 2018). For adeno-associated virus (AAV) vector injection, the mice were deeply anaesthetized with a mixture of ketamine (90 mg kg$^{-1}$; Daiichi Sankyo, Tokyo, Japan) and xylazine (10 mg kg$^{-1}$; Elanco Japan, Tokyo, Japan) and immobilized in a stereotaxic apparatus (SR6N; Narishige, Tokyo, Japan). The scalp skin was cut at the midline, and two craniotomies were performed bilaterally above the LHb [co-ordinates: anteroposterior (AP) −1.4 to −1.7 mm; mediolateral (ML) ±0.5 mm from bregma]. The scalp skin was sutured after removing the glass pipette and cleaning the surgical areas. Mice that received an AAV injection were housed in their home cages for 8 weeks for recovery and induction of *Cnga4* knockdown.

In the experiments shown in Figs. 1*C–E*, *I–K*, 2*G* and 3*B*, three of nine naïve mice (13 of 63 cells), six of nine susceptible mice (33 of 43 cells) and five of nine resilient mice (42 of 55 cells) were examined by investigators who were blinded to their phenotypes. For the depolarizing experiments shown in Figs. 1*A*, *B* and *F–H*, 2*D–F* and 3*C*, three of nine naïve mice (13 of 63 cells), six of nine susceptible mice (33 of 43 cells) and five of nine resilient mice (42 of 55 cells) were examined by investigators who were blinded to their phenotypes. Only male mice were used for the experiments. The investigators understand the ethical principles under which *The Journal of Physiology* operates and certify that our work complies with the animal ethics ARRIVE 2.0 checklist.

## Social defeat stress

Chronic social defeat stress was induced according to a procedure published previously (Golden et al., 2011). Briefly, 7-week-old male C57BL/6J mice were introduced to the home cage of an unfamiliar, aggressive ICR resident mouse for 10 min. Then, the experimental mice were separated from the unfamiliar aggressor by a divider and housed on one side of the cage over the subsequent 24 h. These procedures were repeated for 10 consecutive days. Non-defeated control mice were housed two per cage on either side of a divider.

## Social avoidance test

A social avoidance test was performed by analysing the exploratory behaviour of the mice for the first 2.5 min in an arena (50 × 50 × 40 cm) under illumination with an intensity of 5–20 lux in the presence of a wire mesh enclosure with or without an unfamiliar aggressor mouse. The interaction ratio was calculated as (interaction time, unfamiliar aggressor)/(interaction time, empty cage). The criteria for distinguishing susceptible from resilient mice were defined according to their social interaction behaviour, as reported previously, with modifications (Isingrini et al., 2016). A susceptible mouse was defined by an interaction ratio less than 1 and an interaction time less than 40 s with the aggressor ICR mouse (Table 1). A resilient mouse was defined by an interaction ratio greater than 1 or an interaction time longer than 60 s in the presence of an aggressor. The mice that did not meet the above criteria were classified into the intermediate group.

## Slice preparation

Coronal brain slices (300 μm), including the LHb, were prepared from C57BL6/J mice aged 8–12 weeks. The mice were placed in a chamber, and the CO$_2$ level was subsequently increased. After the mice lost consciousness, they were decapitated. A block of the brain containing the LHb was removed and placed in an ice-cold cutting solution composed of (in mM): 70 sucrose, 75 NaCl, 26 NaHCO$_3$, 2.5 KCl, 5 MgSO$_4$, 2 CaCl$_2$, 1.25 NaH$_2$PO$_4$ and 20 glucose, bubbled with 95% O$_2$ and 5% CO$_2$. Acute brain slices were prepared using a vibrating blade microtome (VT-1200S; Leica Microsystems, Wetzlar, Germany). Slices were incubated for more than 1 h at room temperature in a reservoir with a standard external solution composed of (in mM): 125 NaCl, 2.5 KCl, 2 CaCl$_2$, 1 MgSO$_4$, 1.25 NaH$_2$PO$_4$, 26 NaHCO$_3$ and 20 glucose.

For the experiments for which the results are shown in Figs. 7 and 9, the mice were anaesthetized by an I.P.

injection of a mixture of medetomidine hydrochloride (1.5 mg kg$^{-1}$, Domitor; Nippon Zenyaku Kogyo), midazolam (8 mg kg$^{-1}$, Sandoz; Sandoz) and butorphanol (10 mg kg$^{-1}$, Vetorphale; Meiji Seika Pharma) dissolved in saline. Following complete anaesthesia and analgesia, the mice were perfused transcardially with ice-cold cutting solution containing (in mM): 215 sucrose, 2.5 KCl, 26 NaHCO$_3$, 1.25 NaH$_2$PO$_4$, 20 glucose, 5 sodium pyruvate, 1 CaCl$_2$ and 7 MgCl$_2$ (Velez-Fort et al., 2010) or NMDG solution containing (in mM): 92 NMDG, 2.5 KCl, 1.2 NaH$_2$PO$_4$, 30 NaHCO$_3$, 20 HEPES, 25 glucose, 5 sodium ascorbate, 2 thiourea, 3 sodium pyruvate, 10 MgSO$_4$ and 0.5 CaCl$_2$, titrated to pH 7.3–7.4, using concentrated HCl (Ting et al., 2018). A block of the brain was embedded in 2% agarose and 300 μm coronal slices were prepared. The slices were then incubated for 20 min at 32–34°C in standard external solution or HEPES solution containing (in mM): 92 NaCl, 2.5 KCl, 1.2 NaH$_2$PO$_4$, 30 NaHCO$_3$, 20 HEPES,

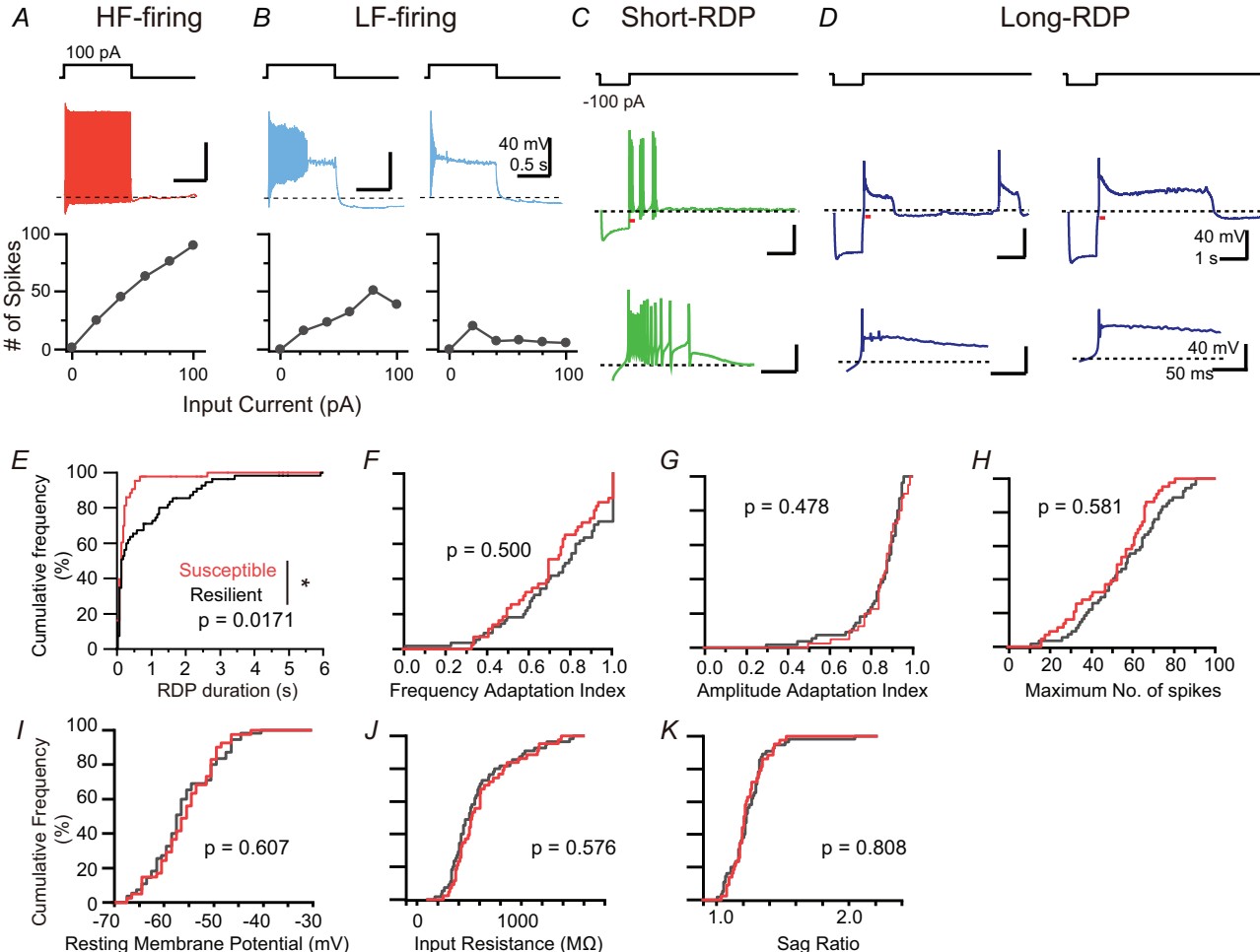

**Figure 1. LHb neurons showing long rebound depolarizing potential responses are decreased in stress-susceptible mice**

*A*, upper: a 100 pA depolarizing current pulse (1 s) applied from the recording electrode; middle: representative spike firings of a HF-firing neuron; Lower: Input current dependence of spike numbers. *B*, representative data from two LF-firing neurons (left and right), similar to those shown in (*A*). *C*, upper: a 100 pA hyperpolarizing current pulse (1 s); middle: a representative voltage trace of a short-RDP in response to a −100 pA hyperpolarizing current injection; lower: the expanded trace from the hyperpolarizing current offset (red line in the middle). *D*, representative traces with medium (left) and long (right) depolarizing phases of RDPs, similar to those shown in (*C*). *E*, cumulative frequency histogram of the RDP durations in response to −100 pA current injections in susceptible (red, *n* = 43, nine mice) and resilient (black, *n* = 55, nine mice) mice. The distributions were significantly different (\**P* = 0.017, Kolmogorov–Smirnov test). *F–H*, cumulative frequency histograms of the frequency adaptation index (*F*), amplitude adaptation index (*G*) and maximum number of spikes (*H*). No distribution was significantly different (Kolmogorov–Smirnov test). *I–K*, cumulative frequency histograms of subthreshold electrophysiological properties (the resting membrane potential (*I*), input resistance (*J*), and voltage sag ratio (*K*)) in susceptible (red, *n* = 43, nine mice) and resilient (black, *n* = 55, nine mice) mice. No distribution was significantly different (Kolmogorov–Smirnov test). [Colour figure can be viewed at wileyonlinelibrary.com]

25 glucose, 5 sodium ascorbate, 2 thiourea, 3 sodium pyruvate, 2 $MgSO_4$ and 2 $CaCl_2$. Thereafter, the slices were incubated for more than 1 h at room temperature. All solutions for slice preparation were bubbled with 95% $O_2$ and 5% $CO_2$.

## Electrophysiology

Whole-cell recordings were performed from visually identified LHb neurons using an upright microscope (BX50WI; Olympus, Tokyo, Japan) equipped with an IR-CCD camera system (C3077-79; Hamamatsu

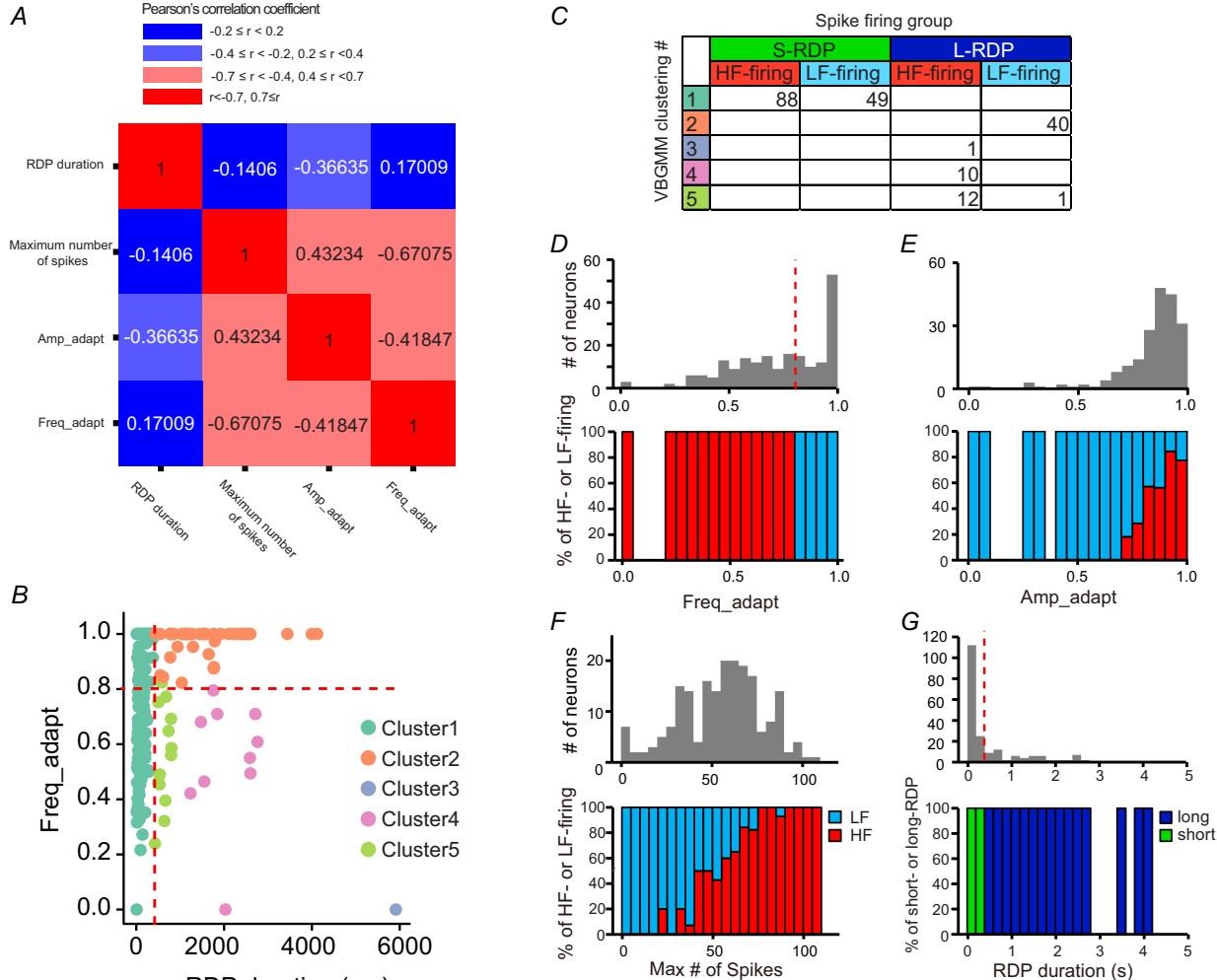

**Figure 2. Classification of lateral habenular neurons based on firing properties**
*A*, correlation matrix of parameters representing firing properties induced by hyperpolarization (RDP duration in response to −100 pA hyperpolarizing currents) and depolarization (maximum number of spikes, frequency adaptation index and amplitude adaptation index). The correlations between parameters representing depolarization-induced spike firing were relatively high. By contrast, the correlations of the RDP duration vs. frequency adaptation index and RDP duration vs. maximum number of spikes were weak. *B*, relationship between the RDP duration and frequency adaptation index ($n = 201$, 44 mice). Data points represent data from individual LHb neurons. The data are colour-coded by clusters determined using a variational Bayesian Gaussian mixture model (VB-GMM). Red and pink lines represent the thresholds of the short- and long-RDP groups and the HF-firing and LF-firing groups, respectively. We used the RDP duration and frequency adaptation index recorded for the social defeat experiment [naïve ($n = 63$, nine mice), susceptible ($n = 43$, nine mice) and resilient ($n = 55$, nine mice) mice] and those used for pharmacological experiments ($n = 40$, 16 mice) to address the changes in the neuronal population induced by the social defeat stress (Table 2). *C*, relationships between clusters by the VB-GMM and groups determined based on thresholds. *D–G*, upper: frequency distribution histograms of the frequency adaptation index (*D*), amplitude adaptation index (*E*), maximum number of spikes (*F*) and RDP duration (*G*); lower: percentages of HF- (red) and LF- (light blue) firing neurons (*D–F*) or short- (green) and long- (blue) RDP neurons (*G*) in individual bars in the frequency distribution histograms. Dotted lines represent the thresholds in (*B*). [Colour figure can be viewed at wileyonlinelibrary.com]

Photonics, Hamamatsu, Japan). The temperature of the extracellular solution was maintained at 32°C. The standard intracellular solution was composed of (in mM): 130 potassium D-gluconate, 10 KCl, 5 NaCl, 0.5 EGTA, 4 Mg-ATP, 0.4 2Na-GTP and 10 HEPES (pH 7.3, adjusted with 1 M KOH). To buffer intracellular $Ca^{2+}$ (Fig. 5*J*), we used an intracellular solution composed of (in mM): 100 potassium D-gluconate, 10 KCl, 5 NaCl, 20 BAPTA tetrapotassium salt, 4 Mg-ATP, 0.4 2Na-GTP and 10 HEPES. The bath was perfused with the standard external solution. The pipette resistance was ∼3–6 MΩ. The membrane potential was recorded under current clamp mode using a Double IPA (Sutter Instruments, Novato, CA, USA), an Axopatch-1D (Molecular Devices, San Jose, CA, USA) or an EPC-10 (Heka Elektronik, Lambrecht, Germany). The signals were filtered at 3 kHz and digitized at 20 kHz. Online data acquisition and off-line data analysis were performed using SutterPatch, version 2.0.0–3.0 (Sutter Instruments), Pulse or Patchmaster, version 2 × 90.2 (Heka Elektronik). The liquid junction potentials were not corrected. Only neurons with a holding current within ±60 pA at a membrane voltage of –60 mV were analysed. In the experiments to analyse the electrophysiological properties of the voltage responses shown in Figs. 1–4, recordings with a series resistance less than 10 MΩ were used for analyses. In pharmacological experiments (Figs. 5–7 and 9), recordings with series resistances less than 20 MΩ were used. In whole-cell

recordings from acute slices prepared from AAV-injected mice, the series resistance tended to be greater than that in other experiments. This greater resistance is probably a result of the poor viability of LHb neurons. For this reason, we also included data from neurons with a series resistance of 10–30 MΩ and those with <10 MΩ for the AAV-injected experiments in Fig. 11*I* and *J*. We confirmed that the RDP duration was less sensitive to the series resistance change [for long-RDP neurons, $R_{series}$ < 12 MΩ, with a value (mean ± SD) of 9.6 ± 0.819 MΩ: 2750 ± 2430 ms, $R_{series}$ > 12 MΩ, with a value of 16.3 ± 2.14 MΩ: 2070 ± 1410 ms ($n = 9$, three mice), $P = 0.910$, Wilcoxon signed-rank test]. In addition, to calculate the cumulative histograms shown in Fig. 11*I* for the AAV-injected mice, the durations of RDPs evoked by the −40 pA or −100 pA hyperpolarizing current were used because neurons were sometimes lost when the −100 pA current was injected. We confirmed that the durations of RDP evoked by −40 pA or −100 pA current injections were identical in control mice (Figs. 5*A* and *B* and 11*I*).

For double whole-cell recording, an intracellular solution with the following composition was used (in mM): 100 potassium D-gluconate, 10 KCl, 5 NaCl, 10 HEPES, 20 EGTA, 4 Mg-ATP and 0.4 2Na-GTP (pH 7.3, adjusted with KOH). A high internal EGTA concentration was used to suppress the closure of gap junctions by the

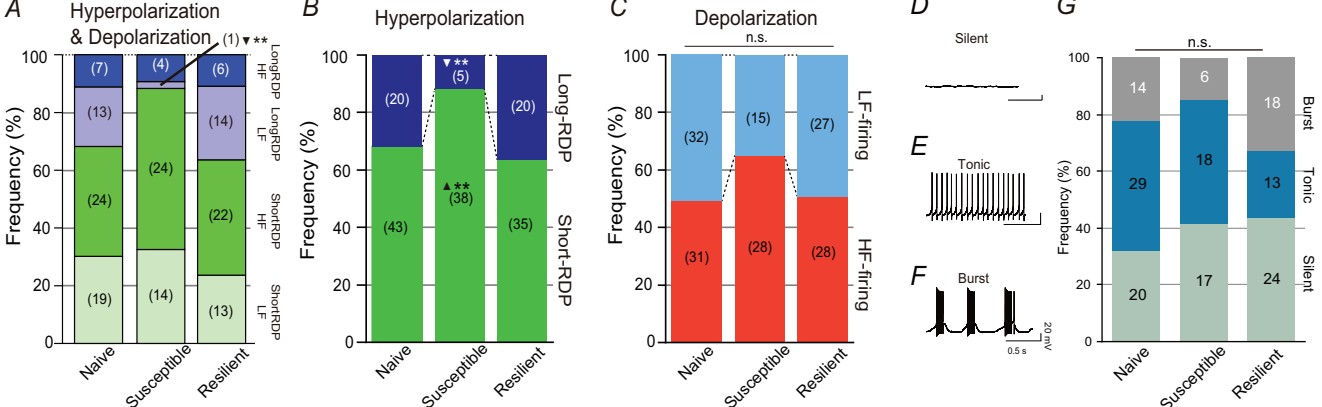

**Figure 3. Stress-induced spike-firing changes of lateral habenular neurons**
*A*, summary of the percentage of short-RDP&HF, short-RDP&LF, long-RDP&HF and long-RDP&LF in naïve ($n = 63$, nine mice), susceptible ($n = 43$, nine mice) and resilient ($n = 55$, nine mice) mice. The numbers in parentheses indicate the number of neurons. Although overall distributions were not statistically significant ($P = 0.0890$, chi-squared), residual analysis determined that long-RDP&LF was significantly reduced in susceptible mice (**$P = 0.00233$; ▼, decrease). *B*, summary of the percentages of the short- and long-RDP neurons among naïve, susceptible and resilient mice. Differences among groups were significant (*$P = 0.0177$, chi-squared). The distribution of susceptible mice was significantly different (**$P = 0.00534$, residual analysis). ▲, increase; ▼, decrease. *C*, summary of the percentages of HF-firing and LF-firing neurons. The differences in distributions were not significant ($P = 0.231$, chi-squared). *D*–*F*, representative traces from LHb neurons exhibiting silent (*D*), tonic (*E*) and burst (*F*) spontaneous firing patterns. In the present study, neurons that were silent or showed firing of less than 0.2 Hz were included in the silent group. Neurons showing both tonic and burst-firing patterns were classified into the burst group. *G*, summary of the percentages of silent, tonic, and burst-firing LHb neurons in naïve ($n = 63$, nine mice), susceptible ($n = 43$, nine mice) and resilient ($n = 55$, nine mice) mice. The difference in the distributions was not significant ($P = 0.0606$, chi-squared). [Colour figure can be viewed at wileyonlinelibrary.com]

**Table 1. Social avoidance test**

| Mouse ID for chronic social defeat stress analysis | Interaction time | Interaction ratio | Phenotype |
|---|---|---|---|
| Mouse 001 | 3 | 0.06 | Susceptible |
| Mouse 002 | 15 | 0.24 | Susceptible |
| Mouse 003 | 22 | 0.43 | Susceptible |
| Mouse 004 | 13 | 0.22 | Susceptible |
| Mouse 005 | 4 | 0.07 | Susceptible |
| Mouse 006 | 18.5 | 0.31 | Susceptible |
| Mouse 007 | 66 | 1.10 | Resilient |
| Mouse 008 | 71 | 1.28 | Resilient |
| Mouse 009 | 21.5 | 0.28 | Susceptible |
| Mouse 010 | 73 | 0.74 | Resilient |
| Mouse 011 | 28.5 | 2.38 | Resilient |
| Mouse 012 | 40.5 | 0.96 | Intermediate |
| Mouse 013 | 9.5 | 0.28 | Susceptible |
| Mouse 014 | 42 | 3.00 | Resilient |
| Mouse 015 | 10 | 0.42 | Susceptible |
| Mouse 016 | 64 | 0.79 | Resilient |
| Mouse 017 | 63.5 | 0.69 | Resilient |
| Mouse 018 | 24 | 1.66 | Resilient |
| Mouse 019 | 58.5 | 1.18 | Resilient |

high internal $Ca^{2+}$ concentration (Bennett et al., 2016; de Mello, 1975; Oliveira-Castro & Loewenstein, 1971). Voltage traces were preprocessed using a low-pass (10 Hz Bessel) filter to remove spike firings and then averaged (4–10 traces). Synchronized voltage changes were assessed by correlation coefficients of voltage traces of pre- and post-junctional cells (Fig. 7*J*).

To measure the RDP duration from traces recorded in the absence of TTX, traces were preprocessed using a low-pass filter to remove spikes. The RDP duration was measured according to the rising and falling phases of the processed RDP when it crossed a threshold of +10 mV above the baseline membrane potential. The RDPs used for analyses of basic electrophysiological properties and clustering (Figs. 1–4) were evoked by hyperpolarizing currents of −100 pA. The RDPs used for pharmacological experiments (Figs. 5, 6 and 7*A–G*) were evoked by hyperpolarizing currents of −40 pA for long and stable recordings. A hyperpolarizing current of −100 pA was used only in the pharmacological experiments shown in Fig. 9 to exclude the possibility that the lack of involvement of CNG channels may be a result of insufficient activation of CNG channels induced by −40 pA current injections. For double whole-cell recording (Fig. 7*H–K*), −100 pA hyperpolarizing currents were used. If neurons were viable, we tested −200 pA hypercurrents to identify clearly the existence of post-junctional responses.

The maximum number of spikes was defined as the maximum number of spikes evoked by a series of 1 s depolarizing current injections from 20 pA to 100 pA in 20 pA steps. The amplitude adaptation index was calculated by dividing the second spike amplitude by the amplitude of the first spike during 100 pA current injection. Peak spike amplitudes from −60 mV were measured. The frequency adaptation index was calculated as:

Frequency adaptation index

$$= \frac{\left|(\text{Initial firing frequency}) - (\text{Final firing frequency})\right|}{\text{Initial firing frequency}},$$

where 'initial firing frequency' was the inverse of the first and second spike intervals and 'final firing frequency' was the average firing frequency during the 200 ms immediately before the offset of depolarization in response to the 100 pA depolarizing current injection.

### Dye coupling

The internal solution was composed of (in mm): 50 potassium D-gluconate, 10 KCl, 20 EGTA, 4 Mg-ATP, 0.4 2Na-GTP, 10 HEPES, 3% neurobiotin (NB) and 0.04% dextran-Alexa Fluor 647 (pH 7.3, adjusted with KOH). Dyes were loaded from the recording electrode for at least 15 min after establishing the whole-cell configuration. The electrode was carefully retracted thereafter, and the acute slice was incubated further for at least 30 min. The acute slices were subsequently immersed in 4% paraformaldehyde and processed for immunostaining. For visualizing NB, slices were washed

and incubated with 0.1% Triton X-100 in PBS for 30 min, followed by incubation in 1:1000 streptavidin-Alexa Fluor 488 (Invitrogen) or 1:500 streptavidin-Alexa Fluor 568 (Invitrogen) for 20 min at room temperature. Representative images were displayed as Z-stacked and subjected to a median filter.

## Clustering using variational Bayesian–Gaussian mixture model

The variational Bayesian–Gaussian mixture model (VB-GMM) was performed using the Python (version 3.10.9) module Scikit-learn package 1.2.1

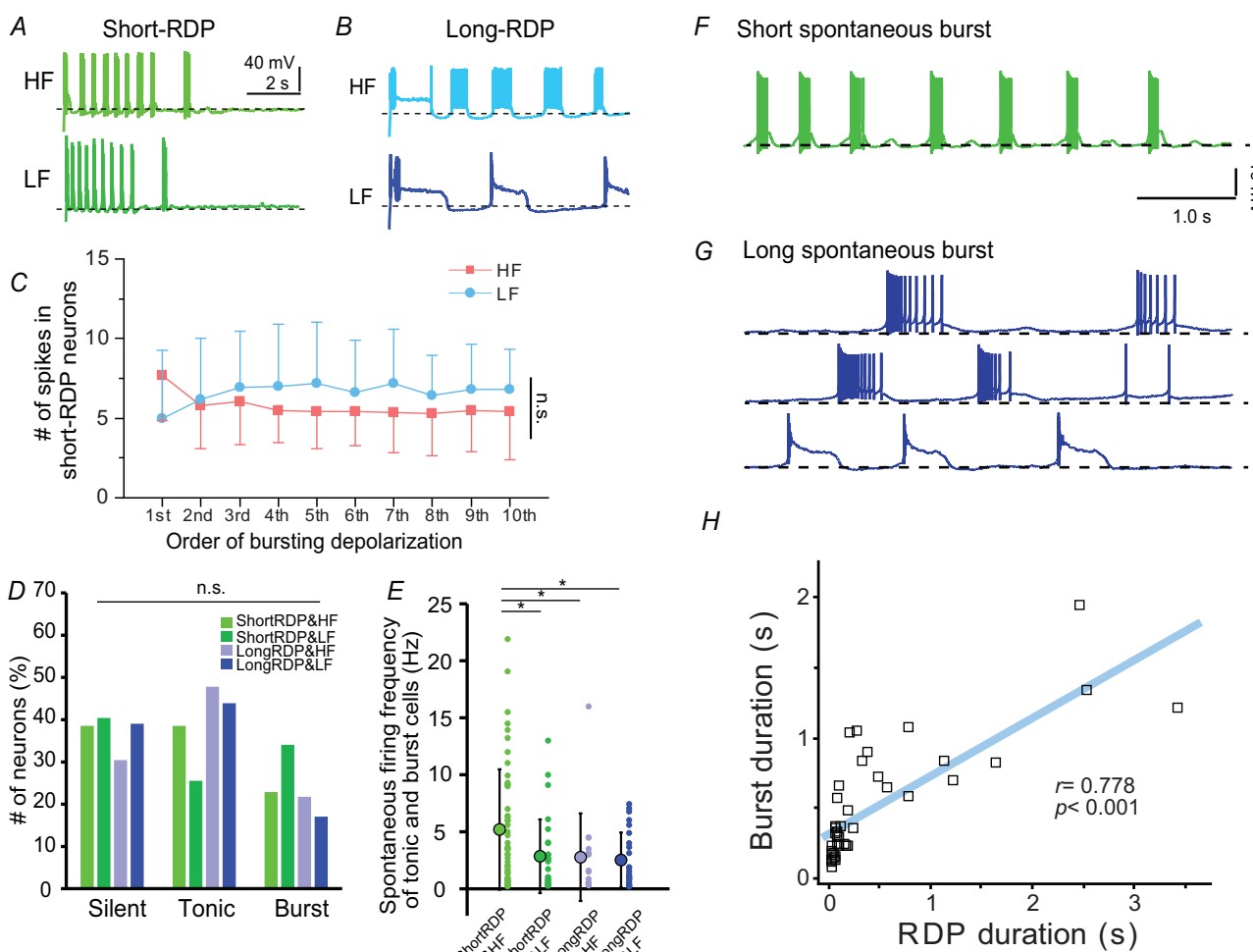

**Figure 4. Evoked and spontaneous firing patterns of LHb neurons**
*A* and *B*, representative traces of short-RDP (*A*) and long-RDP (*B*) neurons with HF-firing (upper) or LF-firing (lower) properties in response to −100 pA hyperpolarizing current injections. *C*, number of spikes that are elicited in the *n*th short bursting depolarizations in short-RDP neurons with HF- (bright red, rectangles, *n* = 15) or LF-firing (light blue, circles, *n* = 11) groups in response to −100 pA hyperpolarizing current injections. Statistical analysis was conducted using two-way repeated measures ANOVA. The effect of cell type (HF- *vs*. LF-firing neurons) was not significant ($F_{1,24}$ = 0.819, *P* = 0.375). *D*, percentages of neurons with silent, tonic and burst-firing patterns in the short-RDP&HF (*n* = 83, 34 mice), short-RDP&LF (*n* = 47, 22 mice, long-RDP&HF (*n* = 23, 17 mice) and long-RDP&LF (*n* = 41, 21 mice) groups. No significant difference was observed (*P* > 0.35, chi-squared). *E*, spontaneous firing rates of tonic and burst neurons in the short-RDP&HF (*n* = 51, 25 mice), short-RDP&LF (*n* = 28, 19 mice), long-RDP&HF (*n* = 16, 13 mice) and long-RDP&LF (*n* = 25, 16 mice) groups. The spontaneous firing rate was calculated by dividing the total number of spikes by their recording time (20 s). The spontaneous firing rate in the short-RDP&HF group was significantly higher than those in other groups ($F_{3,116}$ = 3.48, *P* = 0.0182, one-way ANOVA, **P* < 0.05, Holm–Šidák *post hoc* test was performed between the ShortRDP&LF group and other groups). *F* and *G*, representative traces of short (*F*) and long (*G*) spontaneous depolarizations. Three traces in (*G*) were recorded from different neurons. *H*, relationships between the average durations of spontaneous bursting depolarizations and RDP durations (*n* = 42, 20 mice). A significant correlation was observed (*r* = 0.778, ****P* < 0.001, Pearson correlation coefficient). [Colour figure can be viewed at wileyonlinelibrary.com]

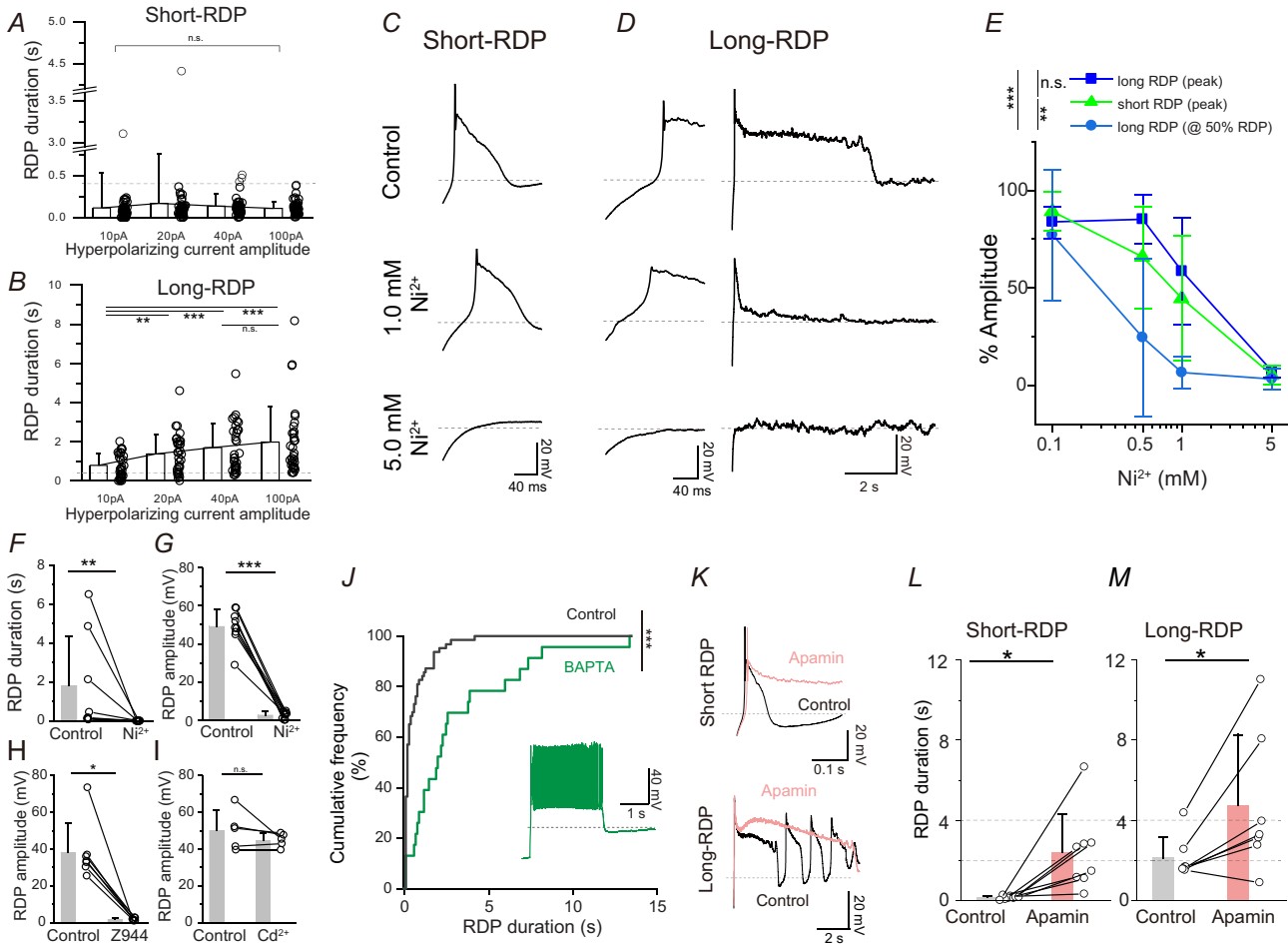

**Figure 5. RDPs are triggered by T-type VDCCs and terminated by SK channels**
*A* and *B*, hyperpolarizing input current dependency of short- (*A*, *n* = 56, eight mice) and long-RDP (*B*, *n* = 32, eight mice) durations. One-way repeated measures ANOVA (short-RDP: $F_{3,165}$ = 0.639, *P* = 0.59; long-RDP: $F_{3,93}$ = 10.54, ****P* < 0.001). Holm–Šidák *post hoc* test for long-RDP (10 pA *vs.* 20 pA: ***P* = 0.00882, 10 pA *vs.* 40 pA: ****P* < 0.001, 10 pA *vs.* 100 pA; ****P* < 0.001, 40 pA *vs.* 100 pA; *P* = 0.213). *C*, representative traces of short-RDPs in the presence of 0 (control), 1, and 5 mM $Ni^{2+}$. Action potentials were blocked by TTX (0.5 µM). *D*, representative traces of long-RDPs. Expanded (left) and long (right) time-scale presentations are shown. The long depolarizing phase was suppressed by the relatively low $Ni^{2+}$ concentration. *E*, dose–response plots of the effects of $Ni^{2+}$ on the peak amplitudes of short-RDPs (green, *n* = 6, five mice), long-RDPs (blue, *n* = 6, five mice) and long depolarizing phases of long-RDPs (pale blue, *n* = 6, five mice). The amplitudes of the long depolarizing phase of long-RDPs were measured at the half RDP duration time point measured in the control solution. Statistical significance was determined using a two-way repeated measures ANOVA [group differences: $F_{2,15}$ = 10.32, *P* = 0.00151], followed by a Holm–Šidák *post hoc* test (short-RDP *vs.* long-RDP (peak); *P* = 0.337, long-RDP (peak) *vs.* long-RDP (@50%RDP); ****P* < 0.001, long-RDP (@50%RDP) *vs.* short-RDP; ***P* = 0.00443]. *F*, suppression of RDP duration by 5 mM $Ni^{2+}$ (*n* = 9, six mice) (***P* = 0.00391, Wilcoxon signed-rank test). *G–I*, suppression of peak RDP amplitudes by 5 mM $Ni^{2+}$ (*G*, *P* < 0.001, RDP amplitude, paired *t* test), 10 µM Z944 (*H*, *n* = 7, two mice, **P* = 0.0156, Wilcoxon signed-rank test) and 100 µM $Cd^{2+}$ (*I*, *n* = 5, three mice, *P* = 0.260, paired *t* test). *J*, cumulative frequency distribution histogram of RDP durations in the presence (green, *n* = 23, two mice) or absence of internal BAPTA (black, *n* = 63, nine mice). Inset: representative RDP trace in the presence of BAPTA. ****P* < 0.001 (Kolmogorov–Smirnov test). *K*, representative short-RDP and long-RDP traces in the presence (red) or absence (black) of apamin (0.1 µM). *L* and *M*, summary of the effects of apamin on short-RDP (*L*, *n* = 8, seven mice) and long-RDP durations (*M*, *n* = 7, five mice). Data from the same neurons are linked. There are significant differences in short- (*P* = 0.0165) and long-RDPs (*P* = 0.0357), according to paired *t* tests. In (*C*) to (*I*) and (*J*) to (*M*), RDPs evoked by hyperpolarizing current injections of −40 pA and −100 pA, respectively, were analysed. [Colour figure can be viewed at wileyonlinelibrary.com]

(sklearn.mixture.BayesianGaussianMixture: https://scikit-learn.org/stable/modules/generated/sklearn.mixture.BayesianGaussianMixture.html). The VB-GMM computes the appropriate posterior distribution over parameters of the GMM under the framework of the VB and automatically defines the number of components of the mixture. The default parameters were used except for the maximum number of mixture components (n_components = 10), the type of the weight concentration prior (weight_concentration_prior_type: 'dirichlet_distribution') and the number of initializations to perform (n_init = 10). Based on correlations between these parameters (Fig. 2*A*), the frequency-adaptation index and RDP duration were used as input data. The estimated number of clusters was always five and the clustering pattern was the same across repeated trials (Fig. 2*B* and *C*). For the input data, we used the RDP duration and frequency-adaptation index of LHb neurons recorded for the social defeat experiment [naïve ($n = 63$, nine mice), susceptible ($n = 43$, nine mice) and resilient ($n = 55$, eight mice) mice] and those used for other experiments involving pharmacological experiments and intermediate type ($n = 40$, 16 mice) to address the neuronal population changes induced by the social defeat stress.

## Analysis of RNA-sequencing data

RNA-sequencing data (available as Series GSE156965 in NCBI Gene Expression Omnibus) previously published as supplementary data (https://static-content.springer.com/esm/art%3A10.1038%2Fs41386-020-00843-0/MediaObjects/41386_2020_843_MOESM9_ESM.xlsx) (Ito et al., 2021) were used in the current analysis. After read count normalization with the trimmed mean of M values normalization (Robinson & Oshlack, 2010), gene expression calculations and statistical analysis based on a generalized linear model were performed using featureCounts (Liao et al., 2014) and the edgeR package (Robinson et al., 2010).

## Immunohistochemistry

The mice were deeply anaesthetized using the same anaesthetic mixture used for the double whole-cell

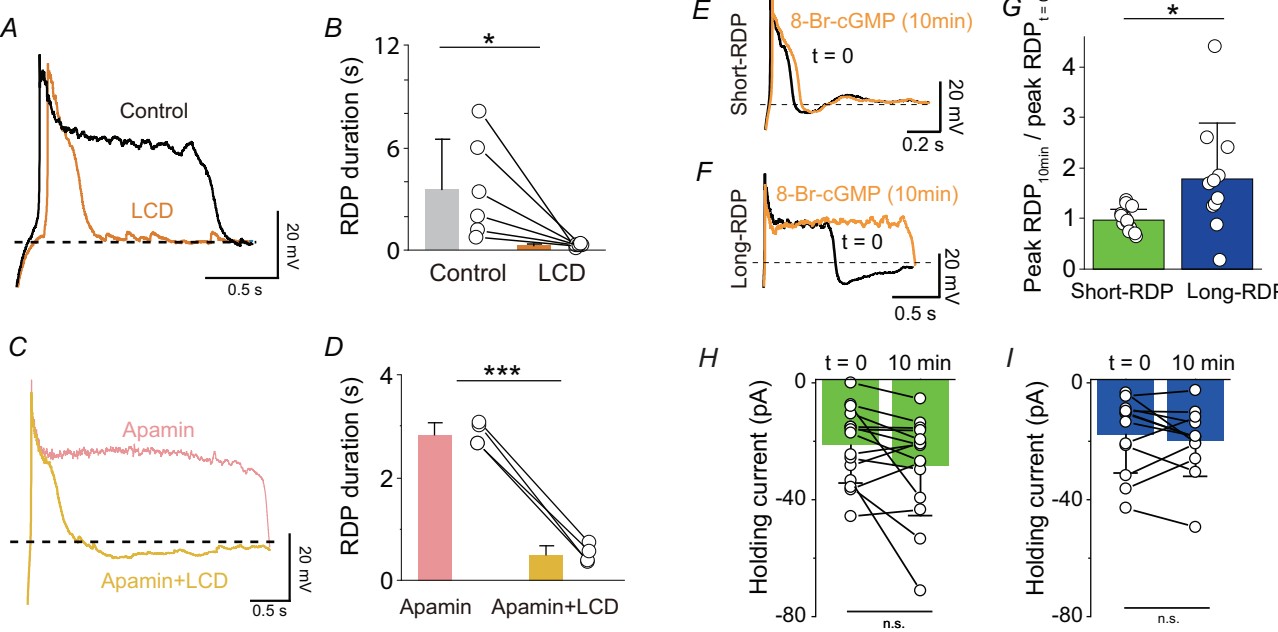

**Figure 6. The long depolarizing phase of long-RDPs is mediated by CNG channels**
*A*, representative long-RDP traces in the presence (orange) or absence (Control, black) of L-*cis*-diltiazem (LCD, 100 μM). *B*, effect of LCD on long-RDP duration ($n = 6$, three mice). *$P = 0.0414$ (paired *t* test). *C*, representative traces showing the effects of LCD in the presence of apamin (0.1 μM). *D*, effect of LCD on RDP duration in the presence of apamin ($n = 4$, three mice ). ****$P < 0.001$ (paired *t* test). *E* and *F*, representative short- (*E*) and long-RDP (*F*) traces in the presence (orange) or absence (black) of internally applied 8-bromo-cGMP (0.5 mM). *G*, summary of the effects of internally applied 8-bromo-cGMP on RDP duration ($n = 25$, three mice). *$P = 0.0344$ (Welch's *t* test). *H* and *I*, effect of internally applied 8-bromo-cGMP on currents necessary to hold the membrane potential at −60 mV for short-RDPs (*H*) and long-RDP (*I*) neurons. There was no significant difference between the groups (paired *t* test, short-RDP: $P = 0.0608$, long-RDP: $P = 0.399$). In (*A*) and (*B*) and (*E*) to (*G*), RDPs evoked by hyperpolarizing current injections of −40 pA were analysed. In (*C*) and (*D*), those evoked by hyperpolarizing current injections of −100 pA were analysed. [Colour figure can be viewed at wileyonlinelibrary.com]

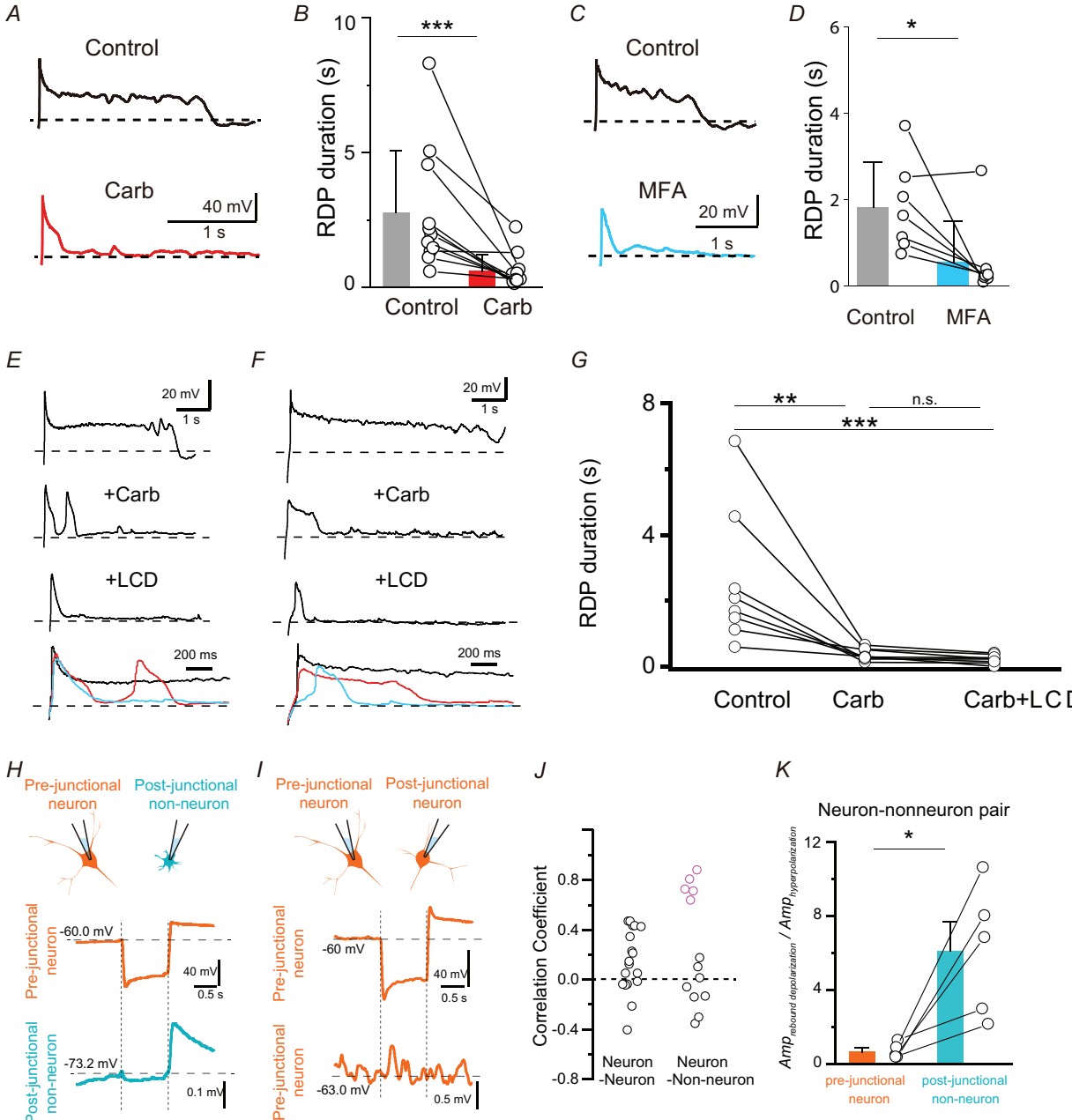

**Figure 7. CNG channels are activated in the post-junctional non-neuronal cells**

*A* and *C*, representative long-RDP traces evoked by the −40 pA hyperpolarizing current injections in the presence [*A*, carbenoxolone (Carb) (red; 100 μM); *C*, meclofenamic acid (MFA) (sky blue; 100 μM)] or absence (black) of gap junction inhibitors. Action potentials were blocked by TTX (0.5 μM). *B* and *D*, effects of carbenoxolone (*B*, n = 11, five mice; ***P < 0.001, Wilcoxon signed-rank test) and MFA (*D*, n = 7, three mice; *P = 0.0369, paired *t* test) on the long-RDP duration. *E* and *F*, representative traces of long-RDPs evoked by −40 pA hyperpolarizing current injections with (*F*) or without (*E*) the pre-junctional response sensitive to L-*cis*-diltiazem. *G*, summary of effects of carbenoxolone and L-*cis*-diltiazem. Statistical analysis was conducted using one-way repeated measures ANOVA ($F_{2,14}$ = 10.80, **P = 0.00145) followed by a Holm–Šidák *post hoc* test (Control *vs.* LCD: ***P < 0.001; Control *vs.* Carb: **P = 0.00163; Carb *vs.* Carb+LCD: P = 0.801). *H*, upper: showing double recording from the neuron–non-neuron pair; lower: representative traces in the pre-junctional neuron and the post-junctional non-neuronal cell in response to −200 pA hyperpolarizing current injection to the pre-junctional neuron. Membrane potentials of neurons were held at approximately −60 mV, but those of non-neuronal cells were at the resting membrane potential. *I*, upper: showing double recording from a neuron–neuron pair; lower: representative traces from the pre-junctional and the post-junctional neurons. *J*, spearman correlation coefficients between traces recorded in pre- and post-junctional cells in response to hyperpolarizing current injections into the

pre-junctional neurons in the neuron–neuron (left, $n = 18$, eight mice) and neuron–non-neuron (right, $n = 13$, eight mice) pairs. Highlighted plots represent non-neuronal cells with rebound depolarization. K, amplitude ratios of the rebound depolarization relative to the preceding hyperpolarization in pre-junctional and post-junctional cells in the neuron–non-neuron ($n = 5$, five mice) pairs with a synchronized response. *$P = 0.0277$ (paired $t$ test). Mouse 019, 58.5 (interaction time), 1.18 (interaction ratio), Resilient (Phenotype). [Colour figure can be viewed at wileyonlinelibrary.com]

**Table 2. The four candidates for a short-guide RNA targeting the *Cnga4* gene**

| No. | Name | Sequence | Strand | PAM | Position in the gene | Cleavage efficiency (%) |
|---|---|---|---|---|---|---|
| 1 | Short guide RNA | | + | CAGAGT | Exon 1 | 58.9 |
| | Forward primer | 5′-CAGGACAGCAAAGTGAAGACAA-3′ | | | | |
| | Reverse primer | 5′-TTGTCTTCACTTTGCTGTCCTG-3′ | | | | |
| 2 | Short guide RNA | | − | ATGGGT | Exon 2 | 82.2 |
| | Forward primer | 5′-CACCAGTAGTAATAATCCCCAG-3′ | | | | |
| | Reverse primer | 5′-CTGGGGATTATTACTACTGGTG-3′ | | | | |
| 3 | Short guide RNA | | + | CGGGAT | Exon 4 | 23.7 |
| | Forward primer | 5′-GTCCAGGTACCTGGGCTTCGGA-3′ | | | | |
| | Reverse primer | 5′-TCCGAAGCCCAGGTACCTGGAC-3′ | | | | |
| 4 | Short guide RNA | | + | CCGAGT | Exon 5 | 53 |
| | Forward primer | 5′-CCGTACACCTGTCTACCCTGAG-3′ | | | | |
| | Reverse primer | 5′-CTCAGGGTAGACAGGTGTACGG-3′ | | | | |

Candidate short-guide RNAs targeting the murine *Cnga4* gene screened in a heteroduplex assay.

experiments. Mouse tissues were fixed by transcardial perfusion with 9% glyoxal and 8% acetic acid (pH 4) after pre-perfusion with 0.9% saline (Konno et al., 2023; Richter et al., 2018). The brain was immersed overnight in the same fixative and then incubated with 30% sucrose in 0.1 M phosphate buffer. Post-fixed brains were embedded in OCT compound (Sakura Finetek, Tokyo, Japan) and frozen. Coronal sections (30–50 μm thick) were cut with a cryostat (Leica Microsystems).

The primary antibodies used were anti-Cnga4 (rabbit, dilution 1:300; Signalway Antibody, College Park, MD, USA; catalog. no. 46522), anti-NeuN (mouse, dilution 1:300; Millipore, Burlington, MA, USA; catalog. no. MAB377), anti-3PGDH (guinea pig, dilution 1:200; Nittobo Medical, Tokyo, Japan; catalog. no. MSFR100030, RRID: AB_2571654), anti-Olig2 (mouse, dilution 1:75; Millipore; catalog. no. MABN50), and anti-HA (mouse, dilution 1:400; BioLegend, San Diego, CA, USA; catalog. no. 901533, RRID:AB_2565005). Slices were incubated with 10% normal donkey serum with 0.1% Triton X-100 in PBS for 30 min, followed by incubation in a mixture of primary antibodies overnight at 4°C and Alexa Fluor 488, Alexa Fluor 568 species-specific secondary antibodies (Invitrogen, Waltham, MA, USA or Jackson Immuno-Research, West Grove, PA, USA) and/or Alexa Fluor 594 (Invitrogen or Jackson ImmunoResearch) secondary antibodies for 2 h at a dilution of 1:500 at room temperature. Images in Figs. 8 and 11 were captured using a STELLARIS 5 confocal microscope (Leica Micro-systems) and a LSM700 confocal microscope (Zeiss,

Oberkochen, Germany), respectively. Figs. 8*B–C*, *F*, *G*, and 11*E–H* were Z-stacked images. For quantitative image analysis, all images were aquired under identical settings and analysed using Fiji software (Schindelin et al., 2012).

## Heteroduplex cleavage assay

The four short-guide RNA candidates (Cnga4#1–4) (Table 2) were transfected into Neuro2A cells in a 24-well plate using Lipofectamine LTX (Life Technologies, Grand Island, NY, USA). At 72 h post-transfection, genomic DNA was isolated using a DNeasy Blood & Tissue Kit (Qiagen, Valencia, CA, USA). The DNA fragments flanking the targeted *Cnga4* loci were then amplified with PCR. For the heteroduplex cleavage assay, the PCR products were denatured, digested at 37°C for 15 min with an EnGen Mutation Detection Kit (E3321S; New England Biolabs, Ipswich, MA, USA), and analysed by electrophoresis in 2% agarose gels stained with ethidium bromide. Gel images were obtained with a Gel Scene GT-33 imager (Astec Co., Ltd, Fukuoka, Japan) and analysed with Fiji software.

To examine the off-target effects of AAV carrying a CRISPR/Cas9 cassette targeting to murine *Cnga4*, we searched its off-target on Benchling (https://benchling.com) based on off target score described previously (Hsu et al., 2013) and found two candidates with 4 or 5 bp mismatches (Table 3). To address whether

the AAV affected these loci, we injected the AAV to the LHb stereotaxically (1.75 mm anterior, 0.6 mm lateral, and 2.8 mm deep to bregma), as described for AAV-based genome editing. The habenular tissues were extracted 2 weeks after AAV transduction. Genomic DNA extracted from the habenula was used as template to PCR-amplified using the following primers: on-target.Fw, 5′-GGAGCTACAGTGCCTCCACCCA-3′;

on-target.Rv, 5′-ACCTGTGTGGAAGCGTACCCCA-3′; off-target1.Fw, 5′-AGCTGCCTAAGGGGAGAAAC-3′; off-target1.Rv, 5′-CCCTTTGGCTGCACATTTAC-3′; off-target2.Fw, 5′- TCTATCAGCTCACTCCCAACG-3′; off-target2.Rv, 5′- AACAAGAGAATTTTCAAGGCAAA-3′. The PCR products were denatured, annealed, and treated by T7 endonuclease I for 60 min at 37°C (catalog.

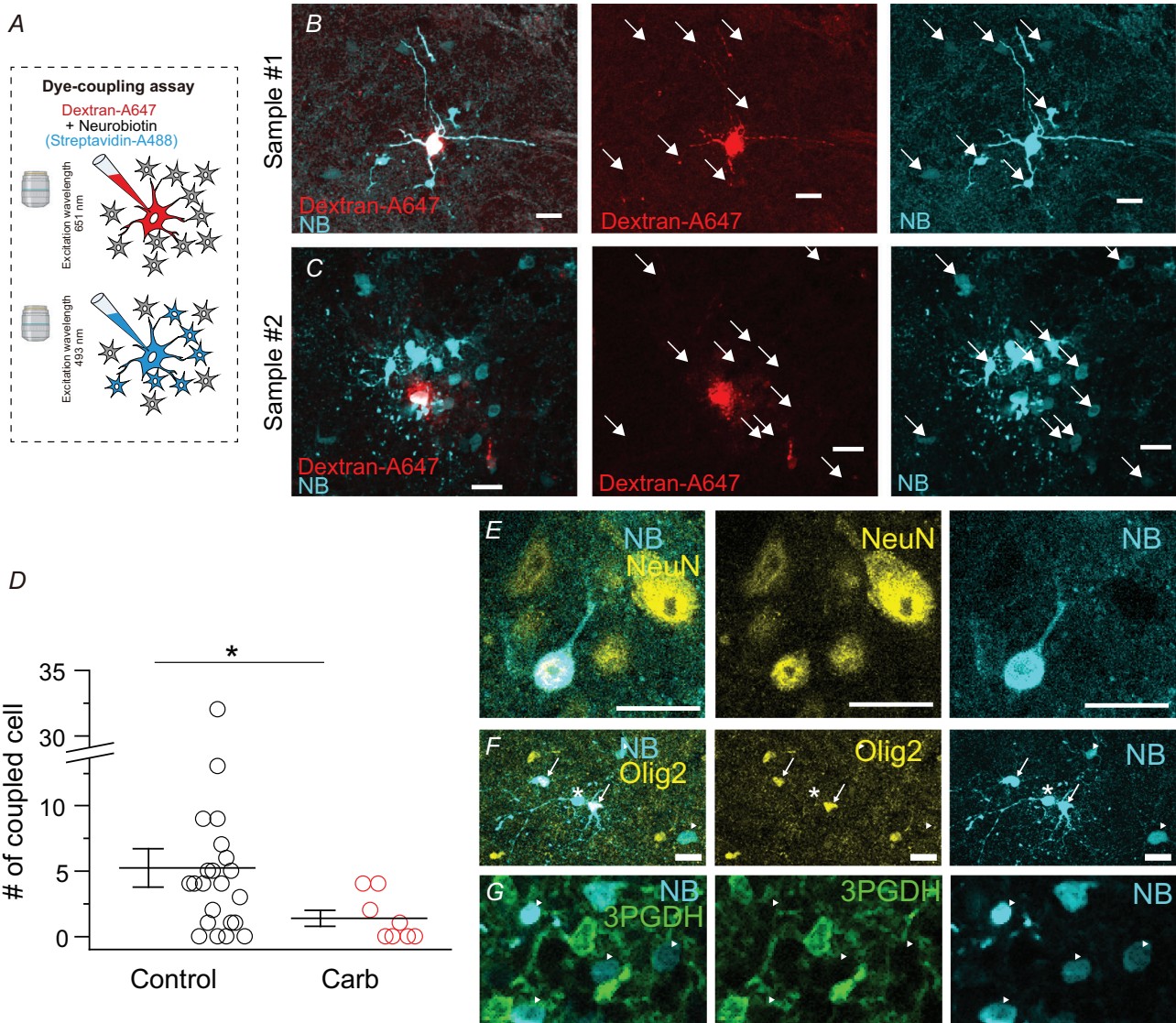

**Figure 8. LHb neurons exhibit dye coupling with neurons and non-neuronal cells, including oligodendrocytes and/or OPCs**

*A*, schema of dye-coupling experiments. *B*, the Z-stack projection image of LHb neurons with dye-coupling cells (white arrows). Gap junction-permeable neurobiotin (NB, cyan) and impermeable dextran-conjugated Alexa Fluor 647 (Dextran-A647, red) were administered via a recording electrode. *C*, another representative image of the dye-coupling experiment. *D*, number of dye-coupled cells per individual LHb neuron in the presence ($n = 8$, four mice) or absence ($n = 22$, nine mice) of carbenoxolone. Averaged data are presented as lines. Carbenoxolone (Carb) significantly reduced the number of dye-coupled cells per individual LHb neuron ($P = 0.0341$, Mann–Whitney *U* test). *E*, NeuN (yellow)-positive dye-coupled cells (cyan). *F*, olig2-positive (arrows) and Olig2-negative (arrowheads) dye-coupled cells (cyan). The recorded neuron is Olig2 negative (asterisk). *G*, 3PGDH-negative (arrowheads) dye-coupled cells (cyan). Scale bars = 20 μm. [Colour figure can be viewed at wileyonlinelibrary.com]

**Table 3. The on-target and two off-target candidate sequences**

|  | Sequence | PAM | Score | Gene | Cut locus |
|---|---|---|---|---|---|
| On target | CACCAGTAGTAATAATCCCCAG | ATGGG | 100.0 | Cnga4 (ENSMUSG00000030897) | chr7:-105 404 918 |
| off target 1 | GTCCAGCAGTAATATTCCACAG | CTGGG | 0.4 |  | chr5:-39 199 076 |
| off target 2 | AACCTGCAGTAATGATCCCCAG | TGGGG | 0.4 |  | chr2:+18 930 870 |

no. 313-08801; Nippon Gene, Toyama, Japan), followed by 3% agarose gel electrophoresis.

### AAV-based genome editing

The CRISPR/Cas9 plasmid was constructed based on a procedure described previously (Cui et al., 2020). Pairs of oligo DNAs corresponding to spacers with adaptors were synthesized, hybridized and ligated using Quick Ligase

(New England BioLabs) into a linearized pX601 plasmid (plasmid #61591; Addgene, Cambridge, MA, USA; Feng Zhang, MIT) digested with *Bsa*I (New England BioLabs). The four candidates, Cnga4#1–#4, for a short-guide RNA targeting the *Cnga4* gene are listed in Table 2. Those and the pCMV-GFP plasmid (plasmid #11 153, Addgene; Connie Cepko, Harvard University), used as a negative control, were transfected into Neuro2A cells in a 24-well plate using Lipofectamine LTX (Life Technologies). At

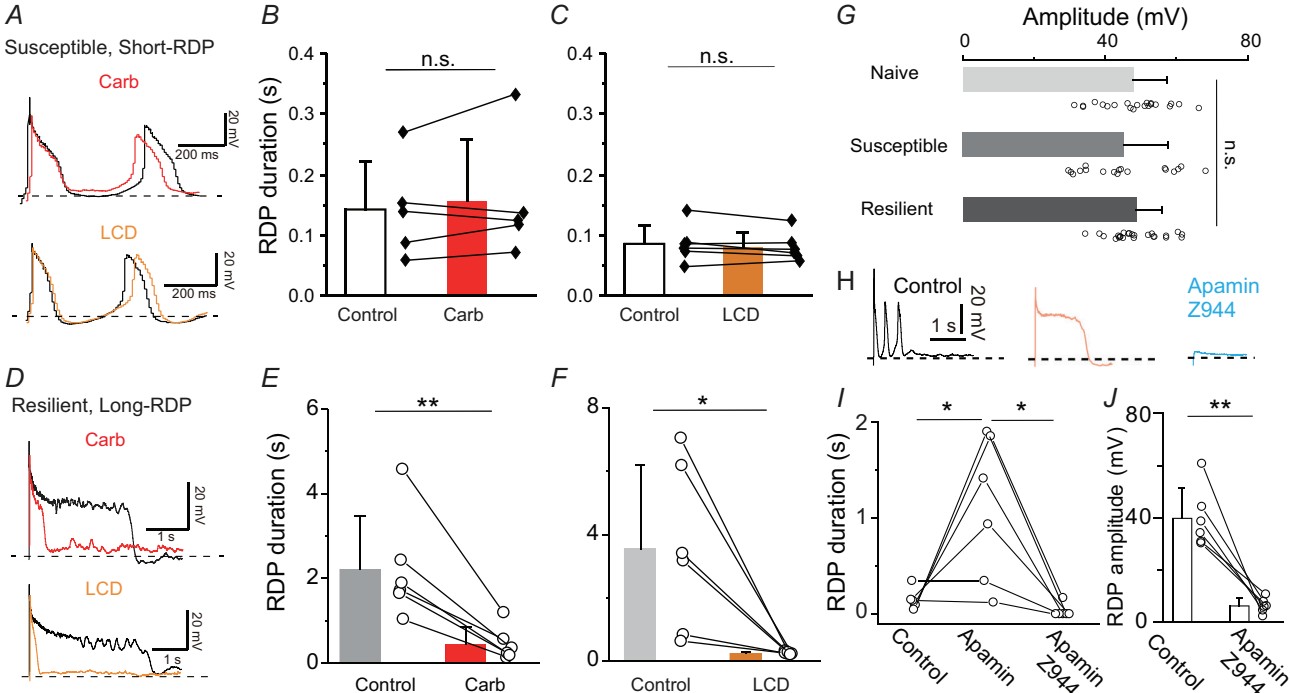

**Figure 9. Short-RDPs in susceptible mice do not involve CNG channel activity**
*A*, representative traces of short-RDPs in the susceptible mice in the presence of 100 μM carbenoxolone (upper, red line) or 100 μM ʟ-*cis*-diltiazem (lower, orange line). Black traces are responses in the control solution. All traces were recorded in the presence of 0.5 μM TTX. *B* and *C*, effects of carbenoxolone (*B*) or ʟ-*cis*-diltiazem (*C*) on short-RDP durations in susceptible mice. There were no significant differences (carbenoxolone; *P* = 0.358, ʟ-*cis*-diltiazem; *P* = 0.327, paired *t* test). *D*, representative traces of long-RDPs in the resilient mice in the presence of 100 μm carbenoxolone (upper, red line) or 100 μM ʟ-*cis*-diltiazem (lower, orange line). *E* and *F*, effects of carbenoxolone (*E*) or ʟ-*cis*-diltiazem (*F*) on long-RDP durations in resilient mice. Long-RDP durations are significantly shortened. \*\**P* = 0.00458 for carbenoxolone, \**P* = 0.0287 for ʟ-*cis*-diltiazem (paired *t* test). *G*, summary of short-RDP amplitudes among the mouse groups. There were no significant differences ($F_{2,60}$ = 0.661, *P* = 0.519, one-way ANOVA). *H*, representative traces of the short-RDP in a susceptible mouse in the control solution with TTX (left, black line) and in the presence of apamin (0.1 μM, middle, red line) or apamin and Z944 (10 μM, right, sky blue). *I*, effects of apamin and apamin and Z944 on short-RDP durations in susceptible mice. \**P* = 0.0384 for Control *vs*. apamin, \**P* = 0.0153 for apamin *vs*. Z944 (paired *t* test). *J*, effects of apamin and Z944 on the RDP amplitude in susceptible mice. \*\**P* = 0.00179 (paired *t* test). All analyses were performed using RDPs evoked by −100 pA hyperpolarizing current injections. [Colour figure can be viewed at wileyonlinelibrary.com]

72 h post-transfection, genomic DNA was isolated using a DNeasy Blood & Tissue Kit (Qiagen). DNA fragments flanking targeted *Cnga4* loci were then amplified from the purified genomic DNA with PCR. For the heteroduplex cleavage assay, the PCR products were denatured, digested at 37°C for 15 min with an EnGen Mutation Detection Kit (E3321S; New England Biolabs), and analysed by electrophoresis in 2% agarose gels stained with ethidium bromide. Gel images were obtained with a Gel Scene GT-33 imager (Astec) and analysed using Fiji software (Schindelin et al., 2012). We chose Cnga4#2, which showed greater cleavage efficiency (Fig. 11*A*).

The AAV vectors for CRISPR/Cas9 were produced and purified as described previously (Sano et al., 2020). In brief, HEK293 cells were co-transfected with a pAAV vector plasmid harbouring a gene of interest, pAAV-DJ and pHelper (catalog. no. VPK-400-DJ; Cell Biolabs, Inc., San Diego, CA, USA). The crude viral lysate was purified by two rounds of caesium chloride ultracentrifugation. The purified viral solution was dialysed with PBS containing 0.001% Pluronic F-68 (Sigma-Aldrich), followed by concentration using an Amicon 10 K MWCO filter (Merck Millipore, Darmstadt, Germany). The viral titre was determined against plasmid standards by real-time PCR using primers that amplified a partial fragment of SaCas9 (5′-AGAAAACAGCAAGAAGGGCA-3′ and 5′-CTTGGCCAGATTCAGGATGT-3′).

For AAV injection, the mice were deeply anaesthetized with a mixture of ketamine (90 mg kg$^{-1}$; Daiichi Sankyo, Tokyo, Japan) and xylazine (10 mg kg$^{-1}$; Elanco Japan, Tokyo, Japan) and immobilized in a stereotaxic apparatus (SR6N; Narishige, Tokyo, Japan). The head skin was cut at the midline, and two craniotomies were performed bilaterally above the LHb (co-ordinates: AP −1.4 to −1.7 mm; ML ±0.5 mm from bregma). The AAV vector was diluted four times with saline and slowly bilaterally injected (80 nL min$^{-1}$) into the LHb (150 nL at each position, four sites; co-ordinates: AP −1.4 mm to −1.7 mm; ML ±0.5 mm; dorsoventral 2.6 mm) through a glass pipette connected to a microsyringe (701SNR; Hamilton, Reno, NV, USA) driven by a pump (YSP-101; YMC, Kyoto, Japan). The glass pipette was held at the injection site for an additional 5 min after injection. After removing the glass pipette and cleaning the surgical areas, the head skin was sutured. The mice that received viral injections were housed in their home cages for 8 weeks for recovery and induction of *Cnga4* knockdown.

## Reagents

Z944 (catalog. no. 6367, CAS 1199236-64-0), apamin (catalog. no. 1652, CAS 24345-16-2), 8-bromo-cGMP (catalog. no. 1089, CAS 51116-01-9) and carbenoxolone disodium (catalog. no. 3096, CAS7421-40-1) were purchased from Tocris Bioscience (Bristol, UK). L-*cis*-Diltiazem (catalog. no. ab120532, CAS42399-54-2) was purchased from Abcam (Cambridge, UK). Tetrodotoxin (catalog. no. 32 775-51, CAS4368-28-9) was purchased from Nacalai Tesque (Kyoto, Japan). BAPTA tetrapotassium salt (catalog. no. B1204), dextran-conjugated Alexa Fluor 647 (catalog. no. D22914, MW 10,000, anionic, fixable) and streptavidin conjugates (Streptavidin Alexa Fluor 488 or 568 conjugate, catalog. no. S11223 or catalog. no. S11226) were purchased from Invitrogen. Neurobiotin (catalog. no. SP-1120) was purchased from Vector Laboratories (Burlingame, CA, USA). Meclofenamic acid (catalog. no. PHR3201, CAS6385-02-0) was purchased from Supelco (Bellefonte, PA, USA).

## Statistical analysis

The data are presented as the mean ± SD, *n* represents the number of LHb neurons and 'mice' represents the number of mice. Comparisons of means between two groups were assessed using *t* tests or Mann–Whitney *U* tests, depending on whether datasets passed normality and equal variance tests. Student's *t* or Welch's *t* test was used, depending on whether the datasets passed the equal variance test. Statistical comparisons of paired samples were assessed using a paired *t* test or a Wilcoxon signed-rank test, depending on whether datasets passed the normality test. Statistical comparisons of means among three or more groups were conducted using one-way ANOVA. When differences were judged significant, data were processed using a Holm–Šidák *post hoc* test. Statistical analysis for Fig. 5*A* and *B* was conducted using a one-way repeated measures ANOVA, whereas a two-way repeated measures ANOVA was used for Figs. 4*C* and 5*E*. Comparisons in Figs. 3 and 4*D* were assessed using a chi-squared test. The Kolmogorov–Smirnov test was used to compare sample distributions between two groups. All tests were two-sided. The data were analysed using Excel (Microsoft Corp., Redmond, WA, USA) and Igor Pro, versions 6–9 (WaveMetrics, Portland, OR, USA). Statistical analyses were conducted using Origin 2018 (OriginLab, Northampton, MA, USA), SigmaPlot, version 12.1 (Systat Software, San Jose, CA, USA) or R, version 4 (R Foundation, Vienna, Austria). $P < 0.05$ was considered statistically significant.

## Results

### Spike-firing patterns of LHb neurons evoked by hyperpolarizing and depolarizing current injections

In the present study, we examined the electrophysiological changes induced by chronic social defeat stress. For this

purpose, we first characterized the depolarization- and hyperpolarization-induced spike-firing properties of LHb neurons using whole-cell recordings from LHb neurons in acute mouse brain slices (see Table S1).

In current clamp mode, we recorded the resting membrane potential and then held the membrane potential at −60 mV to examine spike firing in response to 20 pA depolarizing current steps, ranging from 0 to 100 pA, applied for 1 s. LHb neurons exhibited various evoked firing patterns because of gradual interspike interval extension (frequency adaptation) and/or amplitude reduction (amplitude adaptation) (Fig. 1*A* and *B*) (Weiss & Veh, 2011). These adaptations led to different responses depending on the depolarizing current amplitude: in some neurons, the spike-firing frequency increased monotonically with the depolarizing current (Fig. 1*A*), whereas, in others, the firing frequency was initially increased but later decreased as the depolarizing current amplitude increased (Fig. 1*B*).

We also examined voltage responses to −100 pA hyperpolarizing current injections of 1 s. Some LHb neurons exhibited rebound depolarizing potentials (RDPs) with relatively short durations in response to hyperpolarization (Fig. 1*C*), whereas others displayed RDPs with much longer durations, occasionally lasting several seconds (Fig. 1*D*).

## Neurons with long-RDPs are decreased in mice susceptible to social defeat stress

We investigated how chronic social defeat stress affects spike-firing patterns in LHb neurons (Fig. 1*E–K*). C57BL/6J mice were exposed to chronic social defeat stress from an aggressive ICR mouse. Following the social avoidance test, the mice were classified into 'susceptible' or 'resilient' groups by the social avoidance test. Our procedures categorized ∼50% of the stressed mice as susceptible (Table 1). Non-defeated mice exposed to the same environment without an aggressor were used as 'naïve' mice.

We compared cumulative histograms of various electrophysiological parameters related to the spike-firing evoked by hyperpolarizing and depolarizing current injections across the stressed groups (susceptible and resilient groups) (Fig. 1*E–H*). Notably, we observed significant differences in the RDP duration between the two groups (Fig. 1*E*). Neurons with shorter RDPs were significantly more predominant in stress-susceptible mice than in stress-resilient mice (Fig. 1*E*). By contrast, the electrophysiological parameters related to the spike-firing evoked by depolarizing current injections were not significantly different between the two groups (Fig. 1*F–H*). Similarly, subthreshold electrophysiological parameters (Fig. 1*I–K*) were also comparable between the groups. These results

suggest that the shortening of the RDP duration is closely associated with stress susceptibility.

To explore this further, we classified neurons based on their spike-firing patterns in response to depolarization and hyperpolarization. This classification uses two key parameters: the RDP duration and the frequency-adaptation index (see Methods). RDP duration was used as a parameter to represent hyperpolarization-induced firing (Fig. 1*C* and *D*). For the depolarization-induced firing properties (Fig. 1*A* and *B*), we considered multiple parameters, including the frequency adaptation index, amplitude adaptation index and maximum number of spikes (see Methods). These parameters show relatively high correlations (Fig. 2*A*). Among them, the frequency adaptation index, which exhibited weak correlation with RDP duration (Fig. 2*A*), was chosen as the primary measure of depolarization-induced responses in this study.

Given that both the RDP duration and the frequency adaptation index followed continuous rather than discrete distributions (Fig. 2*D–G*), we applied a VB-GMM to determine thresholds for classification (Kyodo et al., 2023) (Fig. 2*B* and *C*). This analysis revealed that LHb neurons could be grouped into two subgroups based on RDP duration: short-RDP or long-RDP groups at a threshold of 400 ms (Fig. 2*B* and *G*). Similarly, based on the frequency adaptation index, the neurons were classified into two groups: a high frequency firing (HF-firing) group, which showed weak frequency adaptation (frequency adaptation index $\leq 0.8$) and a low frequency firing (LF-firing) group, which exhibited strong frequency adaptation (frequency adaptation index $> 0.8$) (Figs. 1*A* and *B* and 2*D–F*). Compared with LF-firing neurons, HF-firing neurons tended to show lower amplitude adaptation and a greater maximum number of spikes (Fig. 2*D–F*). Combining these classifications, we identified four neuronal subtypes: short-RDP&HF, short-RDP&LF, long-RDP&HF and long-RDP&LF.

We examined the proportions of these four subgroups in naïve, susceptible and resilient mice. Although overall distributions were not statistically significant using the chi-squared test, the residual analysis highlighted a significant reduction in long-RDP&LF-firing neurons in the susceptible mice (Fig. 3*A*). To identify the most crucial electrophysiological properties, we analysed the hyperpolarizing and depolarizing properties separately (Fig. 3*B* and *C*). In naïve mice, although neurons with short-RDP properties were predominant, a significant proportion of neurons exhibited long-RDPs (Fig. 3*B*). In stress-susceptible mice, the proportion of short-RDP neurons was significantly increased, and that of long-RDP neurons was decreased (Fig. 3*B*). By contrast, these changes were not observed in stress-resilient mice (Fig. 3*B*). Meanwhile, the proportions of HF- and LF-firing neurons did not differ significantly among

the naïve, susceptible and resilient groups (Fig. 3*C*). These results are consistent with the findings presented in Fig. 1*E*–*H*, reinforcing the conclusion that a shorter RDP duration is strongly associated with stress susceptibility.

We also examined the influences of stress susceptibility on spontaneous firing. In previous studies, LHb neurons have been classified into three subgroups based on spontaneous firing at the resting potential (Chang & Kim, 2004; Weiss & Veh, 2011; Wilcox et al., 1988). Some neurons are silent at the resting potential (Fig. 3*D*), whereas others exhibit regular or irregular tonic firing (Fig. 3*E*) or repetitive burst firing (Fig. 3*F*). However, we found no significant differences in the proportions of these spontaneous firing patterns between the three groups (Fig. 3*G*).

### An increase in short-RDPs enhances evoked and spontaneous spike firing

We examined their firing dynamics in detail to understand how the increase in short-RDP neurons affects spike firing. A subset of LHb neurons with short-RDPs (68 of 83) exhibited repetitive short depolarizing potentials (1–61 times, $10.39 \pm 11.49$) in an oscillatory pattern within 20 s after the offset of hyperpolarization (Fig. 4*A* and *C*) (Weiss & Veh, 2011; Wilcox et al., 1988). Interestingly, these short bursting potentials were observed even in the LF-firing neurons (Fig. 4*A* and *C*), probably because the brief interburst hyperpolarizations between oscillatory depolarizations allowed recovery from voltage-dependent $Na^+$ channel inactivation. By contrast, spike firing in long-RDP neurons was regular and often became silent during sustained depolarizations, particularly in the LF-firing group (Fig. 4*B*). These data suggest that the increase in short-RDPs enhances spike-firing reproducibility, regardless of whether neurons exhibit HF- or LF-firing properties.

Next, we examined spontaneous firing patterns at the resting potential across the four firing groups. Although the incidences of the silent, tonic and burst groups were similar across the four firing groups (Fig. 4*D*), the overall spontaneous firing rates in the tonic and burst-firing groups were significantly higher in short-RDP&HF neurons (Fig. 4*E*). Furthermore, we observed a significant correlation between the durations of spontaneous firing and RDP (Fig. 4*F*–*H*), which suggests that the evoked RDP and spontaneous response share similar underlying mechanisms. For neurons with LF-firing properties (Fig. 4*G*), long spontaneous depolarizing potentials sometimes appear as prolonged silences of spike firing, during which spike generation is suppressed. These data collectively suggest that the increase in short-RDP neurons enhances both evoked and spontaneous firing

activity, which may contribute to a state of hyper-activation.

### RDPs are triggered by T-type voltage-dependent $Ca^{2+}$ channels (VDCCs) and shortened by small-conductance $Ca^{2+}$-activated $K^+$ (SK) channels

Our data suggested that the RDP duration is closely associated with susceptibility to social defeat stress, and therefore we next examined the ion channels that regulate RDP generation. In subsequent pharmacological experiments, RDPs were recorded in the presence of TTX (0.5 μM) to suppress the action potential firing.

Initially, we investigated the hyperpolarizing prepulse dependence of long-RDPs. The duration of long-RDPs progressively lengthened as the hyperpolarizing current was increased, reaching a plateau at approximately $-40$ pA (Fig. 5*B*). By contrast, the duration of short-RDPs was insensitive to the increase in hyperpolarizing current injections (Fig. 5*A*). These data suggest that the generation of the long depolarizing phase is mediated by factors expressed in long-RDP neurons but not in short-RDP neurons.

Next, we pharmacologically explored the ion channels involved in RDP generation. First, we focused on the role of T-type VDCCs. T-type VDCCs are typically inactivated at the resting membrane potential, but a hyperpolarizing prepulse can promote recovery from inactivation, triggering rebound calcium spikes after the offset of hyperpolarization. The long depolarizing phase of long-RDPs was significantly shortened by the application of a low concentration of $Ni^{2+}$ (0.5–1.0 mM) (Fig. 5*D* and *E*), transforming the response into a short-RDP-like pattern. By contrast, higher concentrations of $Ni^{2+}$ (5 mM) completely blocked both short- and long-RDP responses (Fig. 5*C*–*G*). The observed sensitivity to $Ni^{2+}$ and the requirement for a hyperpolarizing prepulse strongly suggest that both short- and long-RDPs are triggered by rebound calcium spikes mediated by T-type VDCCs. This idea was confirmed by the suppression of both short- and long-RDPs with Z944, a T-type VDCC blocker (Fig. 5*H*). By contrast, $Cd^{2+}$ (0.1 mM), a high-VDCC blocker, did not have a significant effect (Fig. 5*I*). Moreover, these data suggest that the long depolarizing phase of long-RDPs is sensitive to a low concentration of $Ni^{2+}$.

The $Ca^{2+}$ spikes mediated by T-type VDCC activation are usually transient; thus, they alone cannot account for the extremely long duration of long-RDPs. We hypothesized that $Ca^{2+}$ influx might subsequently activate other $Ca^{2+}$-dependent ion channels. To test this hypothesis, we included the $Ca^{2+}$ chelator BAPTA (20 mM) in the internal solution and observed the effects on the RDP duration. Approximately 10 min after establishing whole-cell recordings, the RDP durations

were significantly prolonged (Fig. 5*J*). This finding unexpectedly suggests that $Ca^{2+}$-dependent ion channels might actually shorten RDPs under standard conditions. Further investigation revealed that both short- and long-RDPs, recorded with the standard internal solution, were significantly prolonged by 0.1 μM apamin, a specific blocker of SK channels (Fig. 5*K–M*) (Chang & Kim, 2004), which suggests that SK channels shortened RDPs.

## The long depolarizing phase of long-RDPs is mediated by CNG channels

Although apamin prolonged the duration of short-RDPs, these durations were still shorter than those of long-RDPs (Fig. 5*L* and *M*), suggesting the involvement of other depolarizing ion channels. We focused on $Ni^{2+}$-sensitive ion channels because the long depolarizing phase was shortened by a low concentration of $Ni^{2+}$ (Fig. 5*D* and *E*). Because previous studies have demonstrated that CNG channel activation is affected by $Ni^{2+}$ (Brown et al., 2006; Gordon & Zagotta, 1995; Karpen et al., 1993), we next examined the effects of L-*cis*-diltiazem (100 μM), a CNG channel blocker (Brown et al., 2006). L-*cis*-diltiazem suppressed only the long depolarizing phase of long-RDP responses (Fig. 6*A* and *B*) and converted the response into the short-RDP like response (Fig. 6*A*). Even in the presence of apamin, L-*cis*-diltiazem shortened the long-RDP duration (Fig. 6*C* and *D*). These results suggest that the long depolarizing phase of long-RDPs is mediated by CNG channels. To further examine the contribution of CNG channels, we performed recording from neurons using an intracellular solution containing 8-bromo-cGMP, an analogue of hydrolysis-resistant cGMP that can activate CNG channels. Although 8-bromo-cGMP did not generate a noticeable inward current (Fig. 6*H* and *I*), it significantly prolonged the duration of long-RDPs without affecting short-RDPs (Fig. 6*E–G*), supporting the hypothesis that ion channels that have a sensitivity to cyclic nucleotide contribute to the long depolarizing phase of long-RDPs. These data collectively suggest that the initial large depolarizations of RDPs are driven by rebound $Ca^{2+}$ spikes mediated by T-type VDCCs, followed by activation of CNG channels, which generate the sustained long depolarizing phase. SK channels play a role in shortening RDPs in a $Ca^{2+}$-dependent manner.

## CNG channels are activated via gap junctions

We initially hypothesized that CNG channels were directly activated on LHb neurons. However, we unexpectedly found that the long depolarizing phase of long-RDP was also suppressed by gap junction blockers. Bath-applied carbenoxolone significantly shortened the long-RDP duration (Fig. 7*A* and *B*). A similar result was observed with meclofenamic acid, another inhibitor of gap junctions (Fig. 7*C* and *D*) (Breithausen et al., 2020). These data suggest that CNG channels are activated through gap junctions.

To test this hypothesis, we examined whether L-*cis*-diltiazem would further suppress long-RDPs after the prior application of carbenoxolone. In most LHb neurons (7/11), carbenoxolone largely suppressed the long depolarizing phase of the long-RDPs, converting the response to the short-RDP-like response (Fig. 7*E* and *G*). Subsequent L-*cis*-diltiazem application did not result in any additional suppression, indicating that most CNG channels are predominantly activated in the post-junctional cells. Meanwhile, in a smaller subset of neurons (three of eight), carbenoxolone reduced more than half of the long depolarizing phase, with the remaining depolarization being suppressed by subsequent application of L-*cis*-diltiazem (Fig. 7*F* and *G*). This data suggests that, in these neurons, CNG channels are present on both pre- and post-junctional cells. However, even in these cases, the contribution of pre-junctional CNG channels appeared to be minor compared with that of post-junctional channels (Fig. 7*F* and *G*). These data collectively suggest that post-junctional CNG channels play a central role in generating the long depolarizing phase of long-RDPs in LHb neurons.

We hypothesized that LHb neurons might form electrical coupling with other cells. We conducted double whole-cell recordings from neuron–non-neuronal cell (Fig. 7*H*) or neuron–neuron (Fig. 7*I*) pairs to test this possibility. If electrical coupling was formed, voltage changes induced in the pre-junctional neurons would evoke synchronized voltage responses in the post-junctional cells. We assessed synchronization by calculating the correlation coefficient of voltage traces between the pre- and post-junctional cells (Fig. 7*J*). Non-neuronal cells were identified based on the general electrophysiological properties of glial cells: hyperpolarized resting membrane potential, low input resistance and absence of spikes in response to depolarizing current injection. In the five randomly-sampled neuron–non-neuronal pairs (38.5% (five of 13)) (Fig. 7*H*), hyperpolarizing current injections into the pre-junctional neurons caused synchronized rebound depolarization after the hyperpolarization offset. These pairs of traces showed a relatively high correlation coefficient (Fig. 7*J*). Synchronized hyperpolarization was also observed in these post-junctional non-neuronal cells (Fig. 7*H*), but their amplitudes were much smaller than those in pre-junctional neurons. These smaller amplitudes probably resulted from the lower input resistance of the non-neuronal cells [$R_{neuron} = 529.0 \pm 257$ MΩ, ($n = 16$, eight mice), $R_{non\text{-}neuronal\ cells} = 56.7 \pm 65.0$ MΩ ($n = 13$, eight mice)] and high resistance of gap junctions connecting the

neuron and non-neuronal cells. Consequently, the amplitude ratio of the rebound depolarization relative to that of preceding hyperpolarization was significantly larger in post-junctional non-neuronal cells than in pre-junctional neurons (Fig. 7*K*). These data may suggest that the rebound depolarization was primarily generated in the post-junctional non-neuronal cells. By contrast, no electrical coupling was detected in any of the neuron–neuron pairs in this study (Fig. 7*I* and *J*). These results suggest that electrical coupling is formed between neuron–non-neuronal cell pairs but is minimal, if present, between neuron–neuron pairs.

To identify cell types coupled with LHb neurons, we performed a dye coupling experiment (Fig. 8). Whole-cell recordings from LHb neurons were conducted using an internal solution containing gap junction-permeable NB and impermeable dextran-conjugated Alexa Fluor 647 (Fig. 8*A–C*). The cells were considered NB positive if the average intensity of the cell soma was brighter than the average plus two standard deviations of the intensity of the surrounding area. The cells were considered Alexa Fluor 647 negative if the average intensity of the cell soma was darker than the average plus half of the SD of the intensity of the surrounding area. We excluded NB and Alexa double-positive cells (except the recorded neurons) from the analysis because these were probabaly a result of non-specific staining caused by the internal solution being blown onto nearby cells. In 86.4% (19 of 22) of the LHb neuron recordings, between one and 32 NB-positive and Alexa-negative cells were detected around the recorded LHb neuron (Fig. 8*D*). When carbenoxolone (100 μM) was applied, both the incidence of dye coupling (57.1%, four of seven) and the number of dye-coupled cells was significantly reduced (Fig. 8*D*), suggesting that dye coupling was mediated by gap junctions, at least in part.

To identify coupled cells, some slices were processed for immunostaining using antibodies against NeuN (a marker for neurons), Olig2 (a marker for oligodendrocytes and oligodendrocyte precursor cells or OPCs) and 3PGDH (a marker for astrocytes) (Yamasaki et al., 2001) after dye-coupled cells were stained (Fig. 8*E–G*). Approximately 18.9% of dye-coupled cells were NeuN-positive (seven of 37) (Fig. 8*E*), suggesting that dye coupling was formed among both neuronal and non-neuronal cells but preferentially with non-neuronal cells. A substantial portion of the dye-coupled cells (53.8%, 14 of 26) were labelled with the anti-Olig2 antibody (Fig. 8*F*), suggesting that the dye-coupled cells included oligodendrocytes and/or OPCs. None of the dye-coupled cells were positive for the anti-3PGDH antibody (zero of 45) (Fig. 8*G*), suggesting that the astrocytes were not involved in the coupling. These data suggest that LHb neurons form networks with both neuronal and non-neuronal cells, including oligodendrocytes and OPCs, but not astrocytes.

## CNG channel activity is reduced in susceptible mice

Our data revealed the involvement of CNG channels in long-RDPs, suggesting that a decrease in LHb neurons with long-RDPs in the susceptible mice may be due to decreased CNG channel activity in these mice. To test this possibility, we examined the ion channels responsible for generating short-RDPs in the susceptible mice. As expected, the short-RDPs in the susceptible mice were unaffected by L-*cis*-diltiazem (Fig. 9*A* and *C*) or carbenoxolone (Fig. 9*A* and *B*). Meanwhile, both L-*cis*-diltiazem and carbenoxolone suppressed the long depolarizing phase of long-RDPs in the resilient mice (Fig. 9*D–F*). These data indicate that CNG channels and gap junction networks do not play a role in generating short-RDPs in the susceptible mice.

By contrast, the ion channels responsible for the short-RDP generation were identical in both the control and the susceptible mice, because the short-RDP amplitude was comparable among three mouse groups (Fig. 9*G*) and the short-RDPs in susceptible mice were prolonged by apamin and completely blocked by the subsequent application of Z944 (Fig. 9*H–J*). These data collectively suggest that the increase in short-RDPs in susceptible mice is primarily the result of a reduction in post-junctional CNG channel activity.

## *Cnga4* expression is reduced in susceptible mice

CNG channels have four α-subunits (*Cnga1–4*) and two β-subunits (*Cngb1* and *3*) (Bradley et al., 2005; Brown et al., 2006; Hu & Yang, 2023; Kaupp & Seifert, 2002; Matulef & Zagotta, 2003). We investigated which subtypes contribute to the reduced occurrence of long-RDPs in stress-susceptible mice. We previously acquired data for genes expressed in the LHb of naïve, stress-susceptible and stress-resilient mice using RNA-sequencing (Ito et al., 2021). Our analysis revealed significant *Cnga4* and *Cngb1* expression in the LHb (Fig. 10*A* and *B*). Notably, the *Cnga4* expression was positively correlated with the interaction ratio as a measure of stress susceptibility, i.e. *Cnga4* expression was decreased in the susceptible group compared to that in the resilient group (correlation coefficient, 0.812, $P < 0.001$ for Spearman rank correlation analysis) (Fig. 10*A*). This decreased *Cnga4* expression probably coincides with the decreased long-RDPs in susceptible mice (Figs. 1*E* and 3*A* and *B*). By contrast, the expression of the SK channel and T-type VDCC subtypes was not significantly changed (Fig. 10*C–H*) (Ito et al., 2021). Additionally, the expression or function of hyperpolarization-activated cyclic nucleotide-gated (i.e. HCN) channels, which also generate rebound depolarizations, was unchanged (Fig. 10*I–N*).

We confirmed the contribution of *Cnga4* to the long depolarizing phase of long-RDPs using CRISPR/Cas9

genome editing. We screened four of the short-guide RNA candidates for *Staphylococcus aureus* Cas9 (Cnga4#1–4) (Table 2) and identified that Cnga4#2, which targets exon 2, exhibited the highest cleavage efficiency *in vitro* (Fig. 11*A*). Next, we checked whether Cnga4#2 had any off-target effects by injecting an AAV carrying a Cnga4#2 CRISPR/Cas9 cassette into the LHb and examined two

candidate loci in the genome of AAV-transduced cells predicted by off-target score using heteroduplex cleavage assay (Table 3). The results revealed that Cnga4#2 did not have any apparent off-target effects in these loci, whereas it resulted in the formation of heteroduplex in *Cnga4* locus (Fig. 11*B*), implicating its specificity of genome modification to the *Cnga4* gene. Then, we injected the

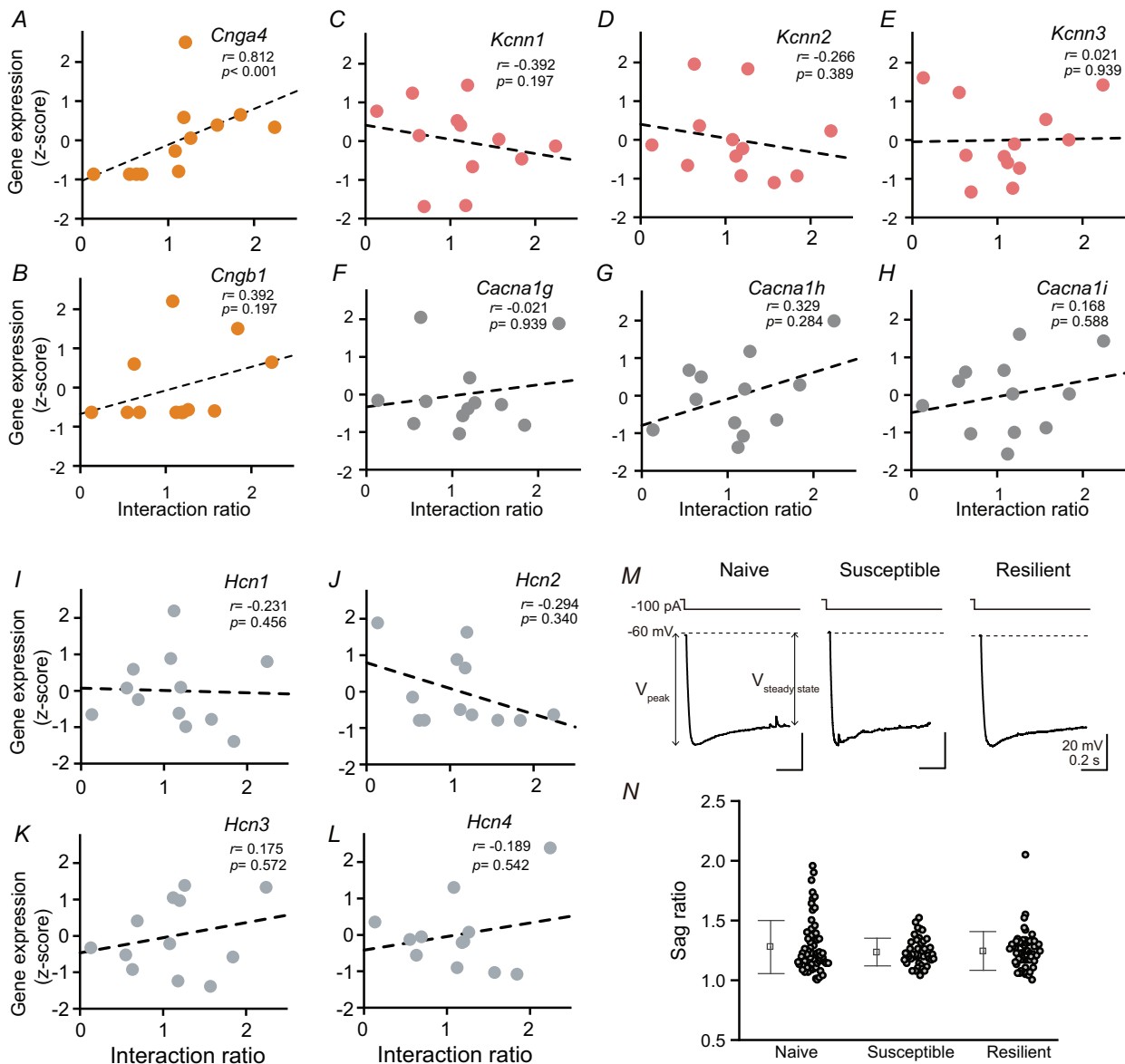

**Figure 10. *Cnga4* expression in the LHb is decreased in susceptible mice**

*A–L*, *Z* scores of gene expression of CNG channels [*Cnga4* (*A*) and *Cngb1* (*B*)], SK channels [*Kcnn1* (*C*), *Kcnn2* (*D*) and *Kcnn3* (*E*)], T-type VDCCs [*Cacna1g* (*F*), *Cacna1h* (*G*) and *Cacna1i* (*H*)] and hyperpolarization-activated cyclic nucleotide-gated channels [*Hcn1* (*I*), *Hcn2* (*J*), *Hcn3* (*K*) and *Hcn4* (*L*)] relative to the interaction ratios. Correlations were assessed using Spearman rank correlation coefficient. *M*, representative voltage traces in response to 100-pA hyperpolarizing current injection in naïve, susceptible, and resilient mice. Most LHb neurons displayed a depolarizing 'sag.' *N*, the depolarizing sag ratio in naïve (n = 69, nine mice), susceptible (*n* = 44, 11 mice) and resilient (*n* = 55, nine mice) mice. The depolarizing sag ratio was calculated as ($V_{peak} - V_{steady state}$)/$V_{peak}$. There was no significant difference among the groups ($F_{2,158}$ = 0.898, *P* = 0.409, one-way ANOVA). [Colour figure can be viewed at wileyonlinelibrary.com]

AAV into the LHb in normal C57BL/6J mice (LHb-*Cnga4* cKO) (Fig. 11*C*). Two months after virus infection, the Cnga4 immunosignal was significantly reduced in the LHb in the LHb-*Cnga4* cKO mice (Fig. 11*D–H*). AAV infection concomitantly shortened the RDP duration in control mice in both the presence (Fig. 11*I*) and absence (Fig. 11*J*) of TTX. The cumulative histogram of RDP

durations in the LHb-*Cnga4* cKO mice closely resembled that in susceptible mice (Fig. 11*J*). Consequently, the percentage of short-RDP neurons in the LHb-*Cnga4* cKO mice increased to the level of that in the susceptible mice (Fig. 11*K*). These data collectively suggest that *Cnga4* is a key subunit responsible for generating the long depolarizing phase of long-duration RDPs.

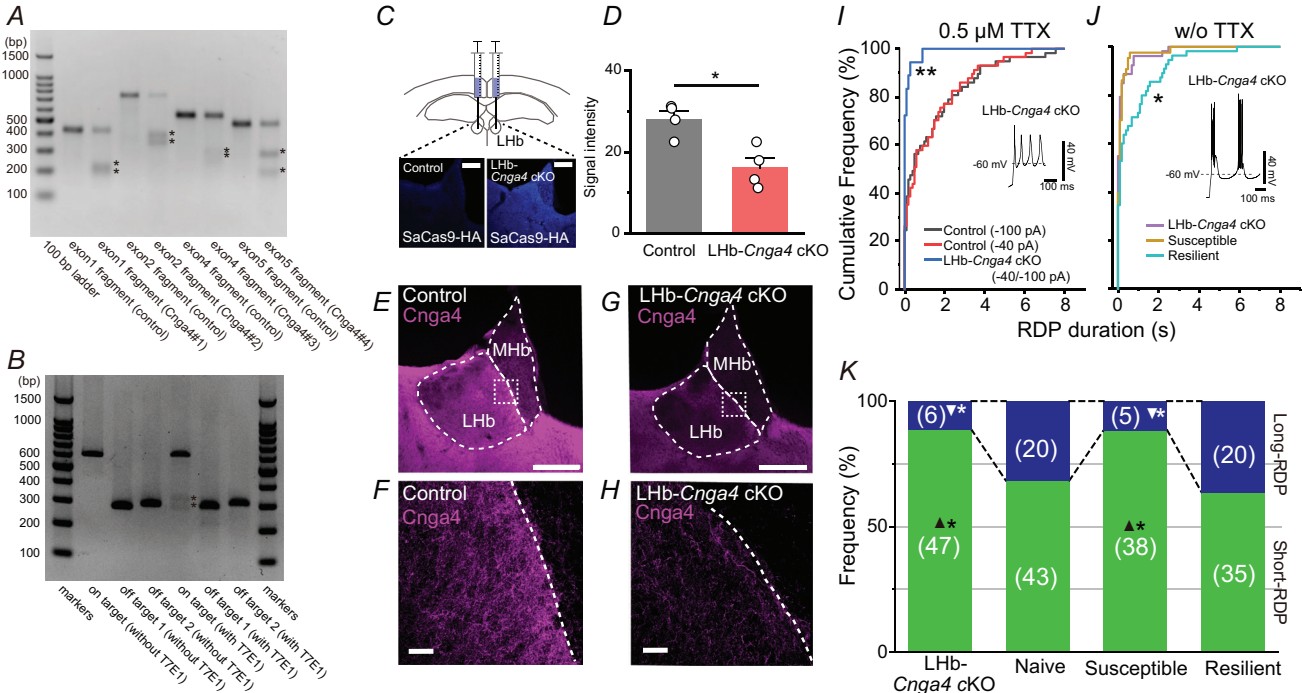

**Figure 11. *Cnga4* is crucial for the generation of the long depolarizing phase of the long-RDP**
*A*, screening to determine an optimal short-guide RNA using a heteroduplex cleavage assay revealed greater cleavage efficiency of the target regions in murine *Cnga4* when short-guide RNA-expressing (Cnga4#1–4) vectors but not the control vector, were transfected into Neuro2A culture cells. *Cleaved fragments. *B*, validation of AAVs carrying CRISPR/Cas9 targeting exon 2 of the *Cnga4* gene *in vivo*. Genomic DNA from the habenula with injection of AAV was used for PCR amplification of the fragment for the Cnga4 gene (lanes 2 and 5), the off-target 1 gene (lanes 3 and 6) and 2 gene (lanes 4 and 7) with (lanes 2–4) and without T7 endonuclease I treatment (lanes 5–7). *Cleaved fragments. *C*, upper: schema of bilateral viral injections of CRISPR adeno-associated virus (AAV) into the LHb; lower: staining of HA-tagged SaCas9 in control (left) and AAV-injected (right) organs involving the LHb. *D*, average intensity of Cnga4 immunosignals in the LHb (*n* = 4, two mice). **P* = 0.0100 (*t* test). *E* and *G*, Cnga4 immunostaining in control (*E*) and AAV-injected (*G*) organs involving the LHb. *F* and *H*, magnified images of the dotted squares in (*E*) and (*G*). *I*, cumulative histograms of RDP duration recorded in the control [*n* = 57, four mice for −40 (red) and −100 pA (black)] and AAV-injected (blue, *n* = 18, four mice for −40 or −100 pA) LHb in the presence of TTX (0.5 μM). (inset) Representative traces of RDPs in the LHb-*Cnga4* cKO mice. Distributions of histograms of RDP durations induced by −40 pA and −100 pA hyperpolarizing current injections are not different. The incidence of short-RDPs significantly increased in LHb-*Cnga4* cKO mice (***P* = 0.00198 for Control-100 *vs*. *Cnga4*-KO-40/100, Kolmogorov–Smirnov test). *J*, cumulative histograms of RDP duration recorded in the AAV-injected (purple, *n* = 53, four mice for −100 pA), susceptible (ochre, *n* = 43, nine mice for −100 pA) and resilient (light green, *n* = 55, nine mice for −100 pA) LHbs in the absence of TTX. (inset) Representative traces of RDPs in the LHb-*Cnga4* cKO mice. The data for the susceptible and resilient mice are the same as those in Fig. 1*E*. The incidence of short-RDPs significantly increased (**P* = 0.0319 for Resilient *vs*. *Cnga4*-KO, Kolmogorov–Smirnov test). *K*, summary of the percentages of short-RDP and long-RDP neurons among mouse groups including AAV-injected one (*n* = 53, four mice). Graphs of naïve (*n* = 63, nine mice), susceptible (*n* = 43, nine mice), and resilient (*n* = 55, nine mice) mice are the same as those presented in Fig. 3*B*. Differences among groups were significant (***P* = 0.00179, chi-squared). The distribution of LHb-*Cnga4*-KO and susceptible mice was significantly different (**P* = 0.0137 and 0.0356, respectively, residual analysis). ▲, increase; ▼, decrease. Scale bars for (*C*), (*E*) and (*G*) = 200 μm. Scale bars for (*F*) and (*H*) = 20 μm. [Colour figure can be viewed at wileyonlinelibrary.com]

## Discussion

### LHb neurons with long-RDPs decrease in the stress-susceptible mice

LHb neurons exhibit considerable morphological and electrophysiological diversity; however, no clear relationship has been observed between their electrophysiological and morphological variability (Kim & Chang, 2005; Weiss & Veh, 2011). It was previously reported that LHb neurons exhibited sustained firing (Weiss & Veh, 2011; Wilcox et al., 1988) or transient firing (Weiss & Veh, 2011) in response to depolarization, which probably corresponds to the HF-firing and LF-firing patterns identified in the present study. Additionally, neurons with short rebound depolarizations (<400 ms) were observed after hyperpolarization (Weiss & Veh, 2011; Wilcox et al., 1988), corresponding to the short-RDPs. Meanwhile, Chang & Kim (2004) reported another subtype of LHb neurons with 'long-lasting depolarizing afterpotentials' following hyperpolarization, which probably corresponds to the long-RDPs in the present study. These lines of evidence suggest that our dataset successfully sampled the diverse LHb neuronal subtypes previously described in the literature.

The present results revealed a marked reduction in the population of long-RDP neurons in stress-susceptible mice, which decreased to less than half the proportion observed in naïve or resilient mice (Figs. 1*E* and 3*B*). In the previous report (Yang et al., 2018), stress-induced changes in the spontaneous spike firing patterns were observed in a subpopulation (~20%) of LHb neurons. Given that the LHb comprises distinct subnuclei with specific efferent targets and functional roles (Aizawa & Zhu, 2019), the stress-induced changes in hyperpolarization-evoked electrophysiological properties may reflect alterations within specific subsets of the LHb neuronal subpopulations. Conversely, depolarization-induced firing properties (proportions of HF- and LF-firing neurons) were not significantly altered by chronic social defeat stress (Fig. 3*C*). This observation may align with the findings of Yang et al. (2018), who demonstrated that repetitive hyperpolarization of LHb neurons using eNpHR3.0 was more effective with respect to inducing depressive behaviours than depolarizations by oChIEF (Yang et al., 2018).

The increase in short-RDP neurons observed in stress-susceptible mice probably contributes to enhancing evoked and spontaneous spike firing within the LHb network (Fig. 4), potentially driving hyperactivation of LHb circuits (Cui et al., 2014; Lecca et al., 2016; Li et al., 2013). Oscillatory depolarizations associated with short-RDPs effectively triggered spike firing even in LF-firing neurons (Fig. 4*A* and *C*). Our data indicate that evoked RDPs and spontaneous firing share similar underlying mechanisms (Fig. 4*H*), suggesting that similar firing alterations would appear in spontaneous responses. Indeed, the spontaneous firing rate was increased in short-RDP neurons (Fig. 4*E*). These data suggest that the increase in short-RDP neurons enhances overall spike firing. By contrast, the spike firing of long-RDP neurons was strongly influenced by their HF- or LF-firing characteristics (Fig. 4*B*). The long depolarization often results in prolonged silences of spike firing, particularly in LF-firing neurons (Fig. 4*B* and *G*), which may lead to reduced activity of these neurons. This property may explain why membrane depolarizations induced by oChIEF fail to effectively evoke high-frequency firing in LHb neurons (Yang et al., 2018). Taken together, our findings suggest that stress-induced changes in RDPs underlie altered spike-firing patterns in chronic stress model animals.

### The long depolarizing phase of long-RDPs is mediated by CNG channels

Both short- and long-RDPs were triggered by rebound $Ca^{2+}$ spikes mediated by T-type VDCCs (Fig. 5*C–H*), which is consistent with previous findings (Yang et al., 2018). The relatively low $Ni^{2+}$ sensitivity (Fig. 5*C–E*) suggests a contribution of the $\alpha$1G (*Cacna1g*) and/or $\alpha$1I (*Cacna1i*) subtypes, which are highly expressed in the LHb (Lee et al., 1999; Wagner et al., 2017). $Ca^{2+}$ influx through these channels subsequently activates SK channels, which act to shorten the duration of RDPs (Fig. 5*K–M*).

Meanwhile, the $Ni^{2+}$ and L-*cis*-diltiazem sensitivities (Figs. 5*C–E* and 6*A* and *B*) indicate that the long depolarizing phase of long-RDPs after $Ca^{2+}$ spikes is mediated by CNG channels. Our data (Figs. 10 and 11) suggest that *Cnga4* and *Cngb1* are expressed in the LHb, and that the reduction in long-RDP neurons observed in susceptible mice can be attributed to the downregulation of *Cnga4* expression (Figs. 9–11). The decreased CNG channel activity may align with our previous findings demonstrating increased expression of the proprotein convertase *Pcsk5* in susceptible mice (Ito et al., 2021). Increased *PCSK5* enhances microglial motility by converting inactive pro-matrix metalloproteinase 14 (pro-MMP14) and pro-MMP2 to their active forms (Ito et al., 2021). MMPs are reported to enhance the ligand sensitivity of CNG channels (*CNGA1/CNGB1* or *CNGA3/CNGB3* heteromeric channels expressed in photoreceptors) (Meighan et al., 2012; Meighan et al., 2013). Furthermore, long-term exposure to MMPs promotes a decrease in active CNG channels (Meighan et al., 2013). Although the specific effects of MMPs on *Cnga4* remain unclear, increased *Pcsk5* activity in susceptible mice might promote decreased *Cnga4* activity,

contributing to the observed increase in short-RDP neurons.

However, it should be noted that both *Cnga4* and *Cngb1* are subunits unable to form homomeric channels activated by cyclic nucleotides (Hu & Yang, 2023; Kaupp & Seifert, 2002; Matulef & Zagotta, 2003). Therefore, their role in promoting the long depolarizing phase remains uncertain, but other CNG subtypes whose expression levels are under the threshold for detection may form functional channels. Alternatively, *Cnga4* might form unique ion channels that are not operated by cyclic nucleotides but by factors unrelated to cyclic nucleotides but triggered by T-type VDCCs or membrane potential depolarization. A previous study reported that gating of the homomeric *CNGA4* (OCNC2) channel can be promoted by nitric oxide (Broillet & Firestein, 1997). Furthermore, even though the channel is not directly activated by cyclic nucleotides (Fig. 6*H* and *I*), the cyclic nucleotides may facilitate its activation (Fig. 6*E*–*G*). Because gap junctions are known to permit the passage of molecules smaller than 1 kDa, 8-bromo-cGMP probably traverses these junctions and facilitates CNG channel activation.

### Neuron–non-neuron networks are formed in the LHb and participate in stress-induced electrophysiological changes of neurons

The present findings indicate that LHb neurons preferentially form functional networks with non-neuronal cells rather than other neurons (Fig. 8). The long depolarizing phase of the long-RDP was probably activated by voltage and/or substrate propagation from the neuron to coupled non-neuronal cells (Fig. 7). Because individual non-neuronal cells exhibited relatively small voltage responses (Fig. 7*H*), collective activity from multiple coupled cells might contribute to the generation of long-RDPs. Among the dye-coupled cells, approximately half were identified as oligodendrocytes and/or OPCs (Fig. 8), suggesting that these cells are strong candidates for expressing *Cnga4*. In the present study, electrical coupling was not observed in the neuron–neuron pairs (Fig. 7*I*), despite a small subset of dye-coupled cells being identified as neurons (Fig. 8*E*). This implies that direct neuron–neuron coupling is rare in the LHb, with neurons instead being indirectly connected through non-neuronal intermediaries. Approximately 30% of dye-coupled cells remain unidentified, indicating the potential diversity and complexity of this network. Similar neuron-non-neuron networks have been reported in other systems. In peripheral sensory ganglia, gap junctions between neurons and satellite glial cells, which correspond to astrocytes in the central nervous system, are upregulated under pathological conditions (Ledda et al., 2009; Spray et al., 2019; Thalakoti et al., 2007). In the central nervous system, gap junctions between neurons and astrocytes have been reported in the locus coeruleus (Alvarez-Maubecin et al., 2000) and cerebellum (Pakhotin & Verkhratsky, 2005). Collectively, these findings suggest that diverse and complex 'pan neuron–non-neuron syncytium' may be widespread and play critical roles in neuronal regulation.

In the LHb, chronic stress has been associated with increased burst spiking, mediated by Kir4.1 upregulation in surrounding astrocytes, which hyperpolarizes the neuronal membrane potential (Cui et al., 2018; Yang et al., 2018). This represents another type of stress-induced neuron–glia interaction that differs from the mechanisms observed in the present study. Here, chronic social defeat stress did not significantly alter the incidence of spontaneous burst firing (Fig. 3*D*–*G*), suggesting a limited role for astrocyte-dependent interactions in this context. However, our data indicate that astrocytes are not involved in dye-coupling networks in the LHb (Fig. 8*G*), which suggests that astrocyte-dependent interactions and non-neuronal network-dependent interactions could function in parallel or independently to mediate stress-induced network alterations. The relative contributions of these mechanisms probably depend on factors such as species, animal models, or the specific stress-induction protocols employed. Together, these findings highlight the complexity and diversity of neuron–glia interactions in regulating LHb network activity under chronic stress.

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

## Additional information

### Data availability statement

All data pertaining to electrical changes induced by social defeat stress (Figs. 1–3) are listed in the Table S1. The datasets generated during and/or analysed during the current study are available from the corresponding author upon reasonable request. The RNA-sequencing data (available as Series GSE156965 in NCBI Gene Expression Omnibus) were previously published as supplementary data (https://static-content. springer.com/esm/art%3A10.1038%2Fs41386-020-00843-0/MediaObjects/41386_2020_843_MOESM9_ESM.xlsx) (Ito et al., 2021).

### Competing interests

The authors declare that they have no competing interests.

### Author contributions

K.Y., H.A. and K.H. were responsible for conceptualization. K.Y., K.N., M.Z., H.T., H.I., M.M., H.T., S.I., Y.S., T.M., H.A. and K.H. were responsible for data curation. K.Y., H.A. and K.H. were responsible for formal analysis. K.Y., K.K., S.I., Y.S., T.M., H.A. and K.H. were responsible for methodology. K.Y., K.N., M.Z., H.T., H.I., M.M., H.T., S.I., Y.S., T.M., H.A. and K.H. were responsible for investigation. K.N., M.Z., H.T., K.K., H.I., M.M., H.T., S.I., Y.S. and H.A. were responsible for resources. K.Y., H.A. and K.H. were responsible for visualization. H.A. and K.H. were responsible for funding acquisition. K.H. was responsible for project administration. K.H. was responsible for supervision. K.Y., K.N., M.Z., H.T., K.K., H.I., M.M., H.T., S.I., Y.S., T.M., H.A. and K.H. were responsible for writing the original draft. K.Y., K.N., M.Z., H.T., K.K., H.I., M.M., H.T., S.I., Y.S., T.M., H.A. and K.H. were responsible for reviewing and editing.

### Funding

This work was supported by the Advanced Research & Development Programs for Medical Innovation (JP23gm6510017) to KH and the Strategic Research Program for Brain Sciences (AMED JP17dm0107093) to HA and KH from AMED. This work was also supported by Grants-in-Aid for Scientific Research (JP19H05723 and 21H02581 to HA, 22K19362 to KH) from the Ministry of Education, Culture, Sports, Science and Technology of Japan.

### Acknowledgements

This work was supported in part by the Natural Science Center for Basic Research and Development (NBARD-00164) in Hiroshima University. We thank Lisa Kreiner, PhD, and Robin James Storer, PhD, from Edanz (https://jp.edanz.com/ac) for editing drafts of this manuscript.

### Authors' present address

H. Ito: Research Facility Center for Science and Technology, Kagawa University, Kagawa, Japan. H. Takemoto: Department

of Physical Therapy, Graduate School of Health and Social Services, Saitama Prefectural University, Saitama, Japan.

## Keywords

cyclic nucleotide-gated channel, dye coupling, gap junction, glia, hyperactivation, lateral habenula, neuron, oligodendrocyte, whole-cell recording

## Supporting information

Additional supporting information can be found online in the Supporting Information section at the end of the HTML view of the article. Supporting information files available:

**Peer Review History**
**Table S1**

