## [Peer Review History · The Journal of Physiology]

Neuron–non-neuron electrical coupling networks are involved in chronic stress-induced electrophysiological changes in lateral habenular neurons

Kenji Yamaoka, Kanako Nozaki, Meina Zhu, Haruhi Terai, Kenta Kobayashi, Hikaru Ito, Miho Matsumata, Hidenori Takemoto, Shinya Ikeda, Yusuke Sotomaru, Tetsuji Mori, Hidenori Aizawa, and Kouichi Hashimoto
DOI: [10.1113/JP287286](https://doi.org/10.1113/JP287286)

Corresponding author(s): Kouichi Hashimoto (hashik@hiroshima-u.ac.jp)

Review Timeline:	Submission Date:	11-Jul-2024
	Editorial Decision:	06-Aug-2024
	Revision Received:	03-Feb-2025
	Accepted:	20-Feb-2025

Senior Editor: Katalin Toth

Reviewing Editor: I-Shan Chen

Transaction Report:

Dear Dr Hashimoto,

Re: JP-RP-2024-287286 "Neuron-nonneuron electrical coupling networks are involved in chronic stress-induced electrophysiological changes in lateral habenular neurons" by Kenji Yamaoka, Kanako Nozaki, Meina Zhu, Haruhi Terai, Kenta Kobayashi, Hikaru Ito, Miho Matsumata, Hidenori Takemoto, Tetsuji Mori, Hidenori Aizawa, and Kouichi Hashimoto

Thank you for submitting your manuscript to The Journal of Physiology. It has been assessed by a Reviewing Editor and by 2 expert referees and we are pleased to tell you that it is potentially acceptable for publication following satisfactory major revision.

LANGUAGE EDITING AND SUPPORT FOR PUBLICATION: If you would like help with English language editing, or other article preparation support, Wiley Editing Services offers expert help, including English Language Editing, as well as translation, manuscript formatting, and figure formatting at www.wileyauthors.com/eoo/preparation. You can also find resources for Preparing Your Article for general guidance about writing and preparing your manuscript at www.wileyauthors.com/eoo/prepresources.

REVISION CHECKLIST:

We look forward to receiving your revised submission.

Yours sincerely,

Katalin Toth
Senior Editor
The Journal of Physiology

REQUIRED ITEMS

- Author photo and profile. First or joint first authors are asked to provide a short biography (no more than 100 words for one author or 150 words in total for joint first authors) and a portrait photograph. These should be uploaded and clearly labelled together in a Word document with the revised version of the manuscript. See Information for Authors for further details.

- Please upload separate high-quality figure files via the submission form.

- You must upload original, uncropped western blot/gel images (including controls) if they are not included in the manuscript. This is to confirm that no inappropriate, unethical or misleading image manipulation has occurred. These should be uploaded as 'Supporting information for review process only'. Please label/highlight the original gels so that we can clearly see which sections/lanes have been used in the manuscript figures. For more information, see: <https://physoc.onlinelibrary.wiley.com/hub/journal-policies#imagmanip>.

- Please ensure that any tables are editable and in Word format, and wherever possible, embedded in the article file itself.

- Please ensure that the Article File you upload is a Word file.

- Papers must comply with the Statistics Policy: https://jp.msubmit.net/cgi-bin/main.plex?form_type=display_requirements#statistics.

In summary:

- If $n \leq 30$, all data points must be plotted in the figure in a way that reveals their range and distribution. A bar graph with data points overlaid, a box and whisker plot or a violin plot (preferably with data points included) are acceptable formats.

- If $n > 30$, then the entire raw dataset must be made available either as supporting information, or hosted on a not-for-profit repository, e.g. FigShare, with access details provided in the manuscript.

- 'n' clearly defined (e.g. x cells from y slices in z animals) in the Methods. Authors should be mindful of pseudoreplication.

- All relevant 'n' values must be clearly stated in the main text, figures and tables.

- The most appropriate summary statistic (e.g. mean or median and standard deviation) must be used. Standard Error of the Mean (SEM) alone is not permitted.

- Exact p values must be stated. Authors must not use 'greater than' or 'less than'. Exact p values must be stated to three significant figures even when 'no statistical significance' is claimed.

- Please include an Abstract Figure file, as well as the Figure Legend text within the main article file. The Abstract Figure is a piece of artwork designed to give readers an immediate understanding of the research and should summarise the main conclusions. If possible, the image should be easily 'readable' from left to right or top to bottom. It should show the physiological relevance of the manuscript so readers can assess the importance and content of its findings. Abstract Figures should not merely recapitulate other figures in the manuscript. Please try to keep the diagram as simple as possible and without superfluous information that may distract from the main conclusion(s). Abstract Figures must be provided by authors no later than the revised manuscript stage and should be uploaded as a separate file during online submission labelled as File Type 'Abstract Figure'. Please also ensure that you include the figure legend in the main article file. All Abstract Figures should be created using BioRender. Authors should use The Journal's premium BioRender account to export high-resolution images. Details on how to use and access the premium account are included as part of this email.

EDITOR COMMENTS

Reviewing Editor:

Comments for Authors to ensure the paper complies with the Statistics Policy (Required):
Data summaries should be presented as mean (SD), not SEM.

Comments to the Author (Required):

Your manuscript has been seen by two expert referees. They both feel that this study is well-constructed and provides novel information of the role of different ion channels associated with RDP. However, several points of critique are raised by referees. Referee #1 suggests additional experiments to address the existence/non-existence of the cAMP/cGMP-sensitive CNG currents in the investigated cells and networks. Referee #2 makes a number of comments for polishing your study.

Please also see 'Required Items' above.

REFEREE COMMENTS

Referee #1:

The authors investigated the electrophysiological mechanisms of the hyperexcitability of glutamatergic LHB neurons. The neurons were classified in 4 groups depending on the responsiveness towards depolarization and/or hyperpolarization, and the authors elucidated the existence of short-RDP/HF-neurons, which may confer the overall hyperexcitability of LHB. Moreover, the mechanisms of the long-RDP were characterized by the coordinated activity of T-type CaV, SK and CNGA4 channels. The possible contribution of CNGA4 in the long-RDP via gap-connected LHB neuron-non-neuron networks in stress-resilience is a very unique and important finding, which is potentially suitable for publication in JP.

However, the proposed idea (CNGA4/b1 subunits contribution without CNGA1/2) is not fully clear: especially the existence/non-existence of the cAMP/cGMP-sensitive CNG currents in the investigated cells and networks. This reviewer has some comments for possible revisions as below.

1)

The non-conductance of CNGA4 homomer and the uniqueness of CNGB homomer are discussed based on the previous literature. Despite the blockers (Ni²⁺, LCD) for the usual CNG currents were used in this study, the actual existence (or non-existence) of the apparent cAMP/cGMP-sensitive CNG currents in the LHB neurons/oligodendrocyte/OPC seems not examined. Voltage-clamp measurement of the inward currents mediated by the cAMP/cGMP from either/both of neurons or non-neuron cells could further reinforce the contribution of the conventional CNG currents in LHB. Alternatively/additionally, the effect (or no effects) of membrane permeable cAMP/cGMP, forskolin, IBMX or etc onto the long-RDP in the I-clamp mode could further elucidate the characteristics of the CNG channel activities in LHB cells.

2)

Please examine the CNGA4-immunosignals with the neuronal/Olig/Astrocyte markers to reinforce the contribution of CNGA4 of oligodendrocyte/OPC is the prime source of rebound potential change.

3)

Fig 9: The representative traces of CNGA4-CRISPR-KO-cells should be helpful to confirm that the short-RDP wave form was unchanged in the KO-cells.

4)

Both T-CaV and conventional CNG channels increase the intracellular Ca^{2+} , thus the feedback via Ca/CaM could affect the SK/CNG activity. In fact, CNGb1 is important in CaM-mediated channel desensitization in olfactory sensory neurons (Neuron. 2008 May 8;58(3):374-86. DOI 10.1016/j.neuron.2008.02.029) and CNGb in LHb might also work as such. Is there any literature for the involvement of Ca/CaM in the long-RDP in LHb or any other stress-dependent neuronal activities?

5)

Fig 1C, D.

A large voltage sag exists in both short-/long-RDP cells, suggesting the existence of HCN channel currents in general. Do you have any information on their expression including subtypes and possible contribution to prolong the RDP?

6)

Stress-induced changes might occur for other ion channels than CNGA4 and their modifiers. Is there any information on the large-scale stress-induced changes in the gene expression in LHb that can account for the stress-susceptibility in addition to CNGA4? Such discussion, if any, may clarify the importance and impact of the CNGA4-mediated long-RDP for stress resilience.

7)

Please provide the information about the off-target effects or any related discussion on possibly affected genes by CRISPR-mediated gene engineering.

8)

Is there any Ca^{2+} -activated Cl^- currents in LHb neurons, which can also change the duration of voltage changes in response to the increase of intracellular Ca^{2+} .

9)

In fig6 E: is there any reason to apply -200 pA injection instead of -100 pA (Fig. 1)?

Referee #2:

In the present study, Yamaoka et al. described neuron-nonneuron electrical coupling networks involved in electrophysiological changes in lateral habenula neurons. The authors showed that the electrophysiological properties of lateral habenula neurons are modulated in mice subjected to social defeat stress. Using pharmacological and genetic approaches, they identified different ion channels involved in short and long RDP, highlighting the cooperation of T-type and SK channels, as well as CNG channels and gap junction. These channels potentially play a role in neuron-nonneuron electrical coupling networks that control RDP in LHb neurons. Beyond these characterizations of RDP in naïve mice, they also demonstrated that in susceptible mice, the proportion of short- and long-RDP neurons changes, supporting the general hypothesis of LHb hyperactivity in stressed mice.

These results are important for the field, as they challenge the classical model of burst activity in the LHb by identifying the role of different ion channels in controlling burst firing. While the study is well-constructed, there are still some experiments and general points to address to support the conclusions and clarify the working model of LHb RDP adaptation after chronic stress.

Major Points:

The authors described the different electrophysiological properties of LHb neurons that adapt according to the state of the animal (susceptible vs. naïve and resilient). The characterization of burst firing induced by hyperpolarization is important because it complements previously published results. The identification of different ion channels involved in LHb neuronal excitability in naïve mice is interesting, yet there is no description of a potential functional alteration of these ion channels in susceptible mice.

- It would be important to characterize, in naïve mice and in a few neurons, the expression of short- and long-RDP depending on the membrane potential of the neurons, as voltage-gated ion channels can also control the duration of the RDP.
- Line 537 "These data suggest that at least a part of CNG channels is activated via the gap." To conclude this, they should prove that there is an occlusion of the LCD effect in the presence of Carb.
- Is there a reduction of T-type, SK, CNG current, or neuron-nonneuron connectivity in susceptible mice compared to resilient and naïve animals?
- To fully support the general conclusions of the paper, it would be important to describe the duration of RDP in susceptible and resilient mice using at least LCD and Carb/MFA blockers. If the general hypothesis is that CNG channels are reduced but not the other channels, we should expect an occlusion of the LCD effect on the duration of the RDP. If the generation of the long-RDP is due to neuron-nonneuron coupling, we could also predict an occlusion of the Carb/MFA effect.

Minor Points:

- The title is not completely appropriate to the findings provided in this study. Indeed, they did not investigate neuron-nonneuron electrical coupling in mice submitted to chronic stress. Yet, if they address the major points, it will be appropriate.
- Figure 1L is not discussed in the results.
- The term "frequency adaptation index" is misleading, as we would expect to have high adaptation properties when close to 1. It would be great to define it better in the results paragraph.
- The description of the four neuron types (HF-short-RDP, HF-long-RDP, LF-short-RDP, LF-long-RDP) is interesting because it takes into account two main electrophysiological properties of LHb neurons. Unfortunately, in Figure 3, the quantification is based only on the duration of RDP or the frequency of the neurons, which raises questions about the pertinence of the cluster analysis. It would be interesting to add the quantification of the neurons in the four groups when comparing naïve, susceptible, and resilient mice.
- Line 532 "Because previous studies have demonstrated that CNG channel activation is affected by Ni²⁺." Considering this, the authors should use a selective blocker of T-type channels (Z944) to really conclude the role of T-type in the generation

of bursts.

END OF COMMENTS

In the present study, Yamaoka et al. described neuron-nonneuron electrical coupling networks involved in electrophysiological changes in lateral habenula neurons. The authors showed that the electrophysiological properties of lateral habenula neurons are modulated in mice subjected to social defeat stress. Using pharmacological and genetic approaches, they identified different ion channels involved in short and long RDP, highlighting the cooperation of T-type and SK channels, as well as CNG channels and gap junction. These channels potentially play a role in neuron-nonneuron electrical coupling networks that control RDP in LHb neurons. Beyond these characterizations of RDP in naïve mice, they also demonstrated that in susceptible mice, the proportion of short- and long-RDP neurons changes, supporting the general hypothesis of LHb hyperactivity in stressed mice. These results are important for the field, as they challenge the classical model of burst activity in the LHb by identifying the role of different ion channels in controlling burst firing. While the study is well-constructed, there are still some experiments and general points to address to support the conclusions and clarify the working model of LHb RDP adaptation after chronic stress.

Major Points:

The authors described the different electrophysiological properties of LHb neurons that adapt according to the state of the animal (susceptible vs. naïve and resilient). The characterization of burst firing induced by hyperpolarization is important because it complements previously published results. The identification of different ion channels involved in LHb neuronal excitability in naïve mice is interesting, yet there is no description of a potential functional alteration of these ion channels in susceptible mice.

- It would be important to characterize, in naïve mice and in a few neurons, the expression of short- and long-RDP depending on the membrane potential of the neurons, as voltage-gated ion channels can also control the duration of the RDP.
- Line 537 "These data suggest that at least a part of CNG channels is activated via the gap." To conclude this, they should prove that there is an occlusion of the LCD effect in the presence of Carb.
- Is there a reduction of T-type, SK, CNG current, or neuron-nonneuron connectivity in susceptible mice compared to resilient and naïve animals?
- To fully support the general conclusions of the paper, it would be important to describe the duration of RDP in susceptible and resilient mice using at least LCD and Carb/MFA blockers. If the general hypothesis is that CNG channels are reduced but not the other channels, we should expect an occlusion of the LCD effect on the duration of the RDP. If the generation of the long-RDP is due to neuron-nonneuron coupling, we could also predict an occlusion of the Carb/MFA effect.

Minor Points:

- The title is not completely appropriate to the findings provided in this study. Indeed, they did not investigate neuron-nonneuron electrical coupling in mice submitted to chronic stress. Yet, if they address the major points, it will be appropriate.
- Figure 1L is not discussed in the results.
- The term "frequency adaptation index" is misleading, as we would expect to have high adaptation properties when close to 1. It would be great to define it better in the results paragraph.
- The description of the four neuron types (HF-short-RDP, HF-long-RDP, LF-short-RDP, LF-long-RDP) is interesting because it takes into account two main electrophysiological properties of LHb neurons. Unfortunately, in Figure 3, the quantification is based only on the duration of RDP or the frequency of the neurons, which raises questions about the pertinence of the cluster analysis. It would be interesting to add the quantification of the neurons in the four groups when comparing naïve, susceptible, and resilient mice.
- Line 532 "Because previous studies have demonstrated that CNG channel activation is affected by Ni²⁺." Considering this, the authors should use a selective blocker of T-type channels (Z944) to really conclude the role of T-type in the generation of bursts.

Comments for Authors to ensure the paper complies with the Statistics Policy (Required):

Data summaries should be presented as mean (SD), not SEM.

[Our response to the comment]

In the revised manuscript, all data summaries are presented as the means \pm SDs.

Comments to the Author (Required):

Your manuscript has been seen by two expert referees. They both feel that this study is well-constructed and provides novel information of the role of different ion channels associated with RDP. However, several points of critique are raised by referees. Referee #1 suggests additional experiments to address the existence/non-existence of the cAMP/cGMP-sensitive CNG currents in the investigated cells and networks. Referee #2 makes a number of comments for polishing your study.

[Our response to the comment]

Thank you for your positive evaluation of our work. We have addressed essentially all the criticisms raised by the reviewers.

Referee #1:

The authors investigated the electrophysiological mechanisms of the hyperexcitability of glutamatergic LHb neurons. The neurons were classified in 4 groups depending on the responsiveness towards depolarization and/or hyperpolarization, and the authors elucidated the existence of short-RDP/HF-neurons, which may confer the overall hyperexcitability of LHb. Moreover, the mechanisms of the long-RDP were characterized by the coordinated activity of T-type CaV, SK and CNGA4 channels. The possible contribution of CNGA4 in the long-RDP via gap-connected LHb neuron-non-neuron networks in stress-resilience is a very unique and important finding, which is potentially suitable for publication in JP.

[Our response to the comment]

We appreciate the reviewer's positive evaluations and helpful suggestions.

1) The non-conductance of CNGA4 homomer and the uniqueness of CNGB homomer are discussed based on the previous literature. Despite the blockers (Ni^{2+} , LCD) for the usual CNG currents were used in this study, the actual existence (or non-existence) of the apparent cAMP/cGMP-sensitive CNG currents in the LHb neurons/oligodendrocyte/OPC seems not examined. Voltage-clamp measurement of the inward currents mediated by the cAMP/cGMP from either/both of neurons or non-neuron cells could further reinforce the contribution of the conventional CNG currents in LHb. Alternatively/additionally, the effect (or no effects) of membrane permeable cAMP/cGMP, forskolin, IMBX or etc onto the long-RDP in the I-clamp mode could further elucidate the characteristics of the CNG channel activities in LHb cells.

[Our response to the comment]

Thank you for your helpful suggestion. We added experiments using 8-bromo-cGMP, a cyclic GMP hydrolysis-resistant analogue, to activate CNG channels. We initially bath-applied 8-bromo-cGMP, but it did not show significant effects on the basal current and long-RDPs recorded in LHb neurons. Therefore, we next applied 8-bromo-cGMP from the recording electrode. Whole-cell recording was performed using a glass microelectrode with the internal solution containing 8-bromo-cGMP (0.5 mM).

Our data revealed that no basal current increase was observed following the establishment of whole-cell recordings (Fig. 6H, I), suggesting minimal ion channel activity was induced by 8-bromo-cGMP in LHb neurons. This lack of current is consistent with previous findings that *Cnga4* and *Cngb1* cannot form homomeric channels directly gated by cyclic nucleotides (Matulef & Zagotta, 2003).

Meanwhile, the duration of long-RDPs was significantly increased by internal 8-bromo-cGMP application (Fig. 6E–G). Because gap junctions are known to permit molecules smaller than 1 kDa to pass through, 8-br-cGMP (*MW* 446.08) likely traversed these junctions and

facilitated CNG channel activation. Our findings suggest that while CNG channels containing *Cnga4* are not directly gated by cyclic nucleotides, their activation may be enhanced in the presence of these molecules. While it is currently unclear how CNG channels are gated, we discussed the possibility that CNG channels involving *Cnga4* may be activated by other factors, such as NO as reported previously (Broillet & Firestein, 1997).

This new data has been included in Figure 6E–I and is now described in the Results section (page 24, line 566–572) and discussed in the Discussion section (page 32, line 751–762).

2) Please examine the CNGA4-immunosignals with the neuronal/Olig/Astrocyte markers to reinforce the contribution of CNGA4 of oligodendrocyte/OPC is the prime source of rebound potential change.

[Our response to the comment]

For immunostaining, we used an anti-Cnga4 antibody (Signalway Antibody Cat# 46522) to check changes in Cnga4 expression in Lhb-*Cnga4* cKO mice (Fig. 11D-H). In higher magnification images, this antibody stained strong spots (arrowheads) and filamentous (arrows) organelles in the Lhb, in addition to weak and cloud-like signals that presented among them (Fig. 1A, B in this response to the reviewers' comments). None of Cnga4 immunosignals did not colocalize with both neuronal (MAP2) (Fig. 1F in this response to the reviewers' comments) and astrocytic (3PGDH) (Fig. 1G in this response to the reviewers' comments) markers. On the other hand, the cloud-like staining of Cnga4 partially colocalized with myelin basic protein (MBP), a marker for oligodendrocytes and oligodendrocyte precursor cells (OPCs) (Fig. 1E in this response to the reviewers' comments), suggesting selective distribution in these cells. However, we noticed that it was difficult to use this immunostaining data for addressing the subcellular distribution of Cnga4.

Fig. 1, Immunostaining using anti-Cnga4 antibody.

(A, C) Cnga4 immunostaining in control (A) and *Cnga4* null (C) mouse organs involving the Lhb. (B, D) Higher magnification images of dotted squares in control (B) and *Cnga4* null (D) mice in A and C, respectively. (E, F, G) Double-immunostaining for Cnga4 and MBP(E), MAP2 (F) and 3PGDH (G).

Given the observed staining patterns, we tried to verify the specificity of these antibodies using *Cnga4* null mice that we generated. The overall immunosignal of Cnga4 in the LHb was decreased in *Cnga4* null mouse (Fig. 1C in this response to the reviewers' comments). Furthermore, the immunosignal was reduced in mice infected with AAV carrying *Cnga4#2* CRISPR/Cas9 cassette (Fig. 11D–H). These data collectively suggest that this antibody can be used to detect overall reduction of Cnga4.

However, in the higher magnification images, we noticed that the strong spot and filamentous staining were not affected in the *Cnga4* null mice (Fig. 1D in this response to the reviewers' comments), suggesting that they are non-specific staining. In contrast, the cloud-like staining was largely absent in the *Cnga4* null mice, suggesting that they reflect Cnga4 distribution. Overall reduction of the Cnga4 signal in *Cnga4* null and LHb-*Cnga4* cKO mice was likely caused by the reduction of these cloud-like signals. Although the cloud-like signals partially overlapped with MBP signals, the presence of relatively strong non-specific staining of this antibody prevents its use for analysis of fine subcellular distributions.

Due to these reasons, we decided to limit the use of this antibody to assessing the overall reduction of Cnga4 in the present study (Fig. 11F–I). In addition, we also tried another antibody (alomone labs Cat# APC-074), but found that its immunosignals were not affected in *Cnga4* null mice (data not shown).

3) Fig 9: The representative traces of CNGA4-CRISPR-KO-cells should be helpful to confirm that the short-RDP wave form was unchanged in the KO-cells.

[Our response to the comment]

We have added representative traces of RDPs recorded from LHb-selective *Cnga4* knockout mice in Fig. 11I, J.

4) Both T-CaV and conventional CNG channels increase the intracellular Ca^{2+} , the feedback via Ca/CaM could affect the SK/CNG activity. In fact, CNGb1 is important in CaM-mediated channel desensitization in olfactory sensory neurons (Neuron. 2008 May 8;58(3):374-86. DOI 10.1016/j.neuron.2008.02.029) and CNGb in LHb might also work as such. Is there any literature for the involvement of Ca/CaM in the long-RDP in LHb or any other stress-dependent neuronal activities?

[Our response to the comment]

We have conducted additional experiments to investigate the involvement of CaMs in the generation of RDPs. However, bath application of W-7 hydrochloride (100 μ M), a CaM blocker, did not affect

the duration of long-RDPs (control: 2778 ± 846.3 ms, W-7 hydrochloride: 2693 ± 1598 ms, (n = 6, mice = 3), $p = 0.909$, paired t test). This data suggests that the Ca^{2+} /CaM signalling cascade is not likely involved in the long-RDP generation.

5) Fig 1C, D. A large voltage sag exists in both short-/long-RDP cells, suggesting the existence of HCN channel currents in general. Do you have any information on their expression including subtypes and possible contribution to prolong the RDP?

[Our response to the comment]

We have now included gene expression data for HCN channel subtypes across the naïve, susceptible, and resilient mice in Fig. 10I–L. Additionally, we analysed changes in voltage sags among mouse groups (Fig. 10M, N). Our findings revealed that both the expression and function of HCN channels remain unaffected by chronic social defeat stress.

This new data has been included in Fig. 10I–N and is now described in the Results section (page 28, lines 670–672).

6) Stress-induced changes might occur for other ion channels than CNGA4 and their modifiers. Is there any information on the large-scale stress-induced changes in the gene expression in LHb that can account for the stress-susceptibility in addition to CNGA4? Such discussion, if any, may clarify the importance and impact of the CNGA4-mediated long-RDP for stress resilience.

[Our response to the comment]

According to our previous RNA-seq data (Ito *et al.*, 2021), differentially expressed genes analysis comparing control, susceptible, and resilient groups revealed that *Cnga4* was the top-ranked ion channel component gene based on p -value and FDR ($***p < 0.001$, FDR = 0.00665). This finding highlights the important role of *Cnga4* in the mechanisms underlying stress vulnerability.

Regarding modifiers, we have now incorporated the following discussion to explain the stress-induced changes in the gene expression based on our previous research on page 31, line 742–page 32, line 750.

The decreased CNG channel activity may align with our previous findings demonstrating increased expression of the proprotein convertase *Pcsk5* in susceptible mice (Ito *et al.*, 2021). Increased *PCSK5* enhances microglial motility by converting inactive pro-matrix metalloproteinase 14 (pro-MMP14) and pro-MMP2 to their active forms (Ito *et al.*, 2021). MMPs are reported to enhance the ligand sensitivity of CNG channels (*CNGA1/CNGB1* or *CNGA3/CNGB3* heteromeric channels expressed in photoreceptors) (Meighan *et al.*, 2012; Meighan *et al.*, 2013). Furthermore, long-term exposure to MMPs promotes a decrease in active CNG channels (Meighan *et al.*, 2013). Although the specific effects of MMPs on *Cnga4* remain unclear, increased *Pcsk5* activity in susceptible mice might promote decreased *Cnga4* activity, contributing to the observed increase in

short-RDP neurons.

7) Please provide the information about the off-target effects or any related discussion on possibly affected genes by CRISPR-mediated gene engineering.

[Our response to the comment]

We searched for off-targets on the Benchling based on off target score and found two candidates with 4-5 base pair mismatches (Table 3). The habenular tissue sampled from AAV-injected mice was processed for the heteroduplex cleavage assay to check off-target effects. Genomic DNA extracted from the habenula was used as a template for PCR. We found that AAV infection cleaved some *Cnga4* genome (* in Fig. 11B) but not two off-target candidates (Fig. 11B). These data suggest that *Cnga4* was preferentially cleaved by AAV injection in our experiments.

This new data is now shown in Fig. 11B and Table 3, and described in the Result section (page 29, line 676–681)

8) Is there any Ca^{2+} -activated Cl^- currents in LHb neurons, which can also change the duration of voltage changes in response to the increase of intracellular Ca^{2+} .

[Our response to the comment]

A previous study reported that hyperpolarization of LHb neurons elicited ‘long-lasting depolarizing after potential (LDAP)’ (Chang & Kim, 2004), which closely resembles the long-RDPs described in our study. The authors reported that LDAP was suppressed by flufenamic acid, a blocker for non-specific cationic currents. Because flufenamic acid is also known to suppress Ca^{2+} -activated Cl^- currents (White & Aylwin, 1990; Oh *et al.*, 2008), we tested the effect of flufenamic acid on the long-RDPs. Our results show that the 50 μM flufenamic acid did not suppress the duration of the long-RDPs (control: 5270 ± 3006 ms, flufenamic acid: 4977 ± 4552 ms, ($n = 5$, mice=2), $p = 0.849$, paired t test).

Furthermore, we investigated differences in expression of Ca^{2+} -activated Cl^- channels among naive, susceptible and resilient mouse groups. According to our data reported by Ito *et al.* (Ito *et al.*, 2021), genes relating to Ca^{2+} -activated Cl^- channels, including *Ano1*, *Clca3a1*, *Ano6*, *Clca4a*, and *Kcmb1* were detected in the LHb. We found that their differential expression analysis comparing control, susceptible, and resilient groups revealed no statistically significant changes in their expression (*Ano1* [$p = 0.622$, FDR > 0.999], *Clca3a1* [$p = 0.330$, FDR > 0.999], *Ano6* [$p = 0.796$, FDR > 0.999], *Clca4a* [$p = 0.930$, FDR > 0.999], *Kcmb1* [$p = 0.657$, FDR > 0.999]). Taken together, these findings suggest that Ca^{2+} -activated Cl^- channels do not play a significant role in generating long-RDPs.

Taken together, these findings suggest that Ca^{2+} -activated Cl^- channels do not play a significant role in generating long-RDPs.

9) In fig6 E: is there any reason to apply -200 pA injection instead of -100 pA (Fig. 1)?

[Our response to the comment]

In most of our experiments, we applied a hyperpolarizing current of -40 or -100 pA. However, this current did not elicit any responses in the post-junctional neurons (Fig. 7I). To further confirm the absence of post-junctional potentials, we increased the pre-junctional input current to -200 pA only in the double recording experiment.

Explanation for hyperpolarizing currents used in the present analysis is now presented in the Methods section (page 12, lines 256–263).

Referee #2:

In the present study, Yamaoka et al. described neuron-nonneuron electrical coupling networks involved in electrophysiological changes in lateral habenula neurons. The authors showed that the electrophysiological properties of lateral habenula neurons are modulated in mice subjected to social defeat stress. Using pharmacological and genetic approaches, they identified different ion channels involved in short and long RDP, highlighting the cooperation of T-type and SK channels, as well as CNG channels and gap junction. These channels potentially play a role in neuron-nonneuron electrical coupling networks that control RDP in Lhb neurons. Beyond these characterizations of RDP in naïve mice, they also demonstrated that in susceptible mice, the proportion of short- and long-RDP neurons changes, supporting the general hypothesis of Lhb hyperactivity in stressed mice.

These results are important for the field, as they challenge the classical model of burst activity in the Lhb by identifying the role of different ion channels in controlling burst firing. While the study is well-constructed, there are still some experiments and general points to address to support the conclusions and clarify the working model of Lhb RDP adaptation after chronic stress.

[Our response to the comment]

We appreciate the reviewer's positive evaluations and helpful suggestions.

Major Points:

The authors described the different electrophysiological properties of Lhb neurons that adapt according to the state of the animal (susceptible vs. naïve and resilient). The characterization of burst firing induced by hyperpolarization is important because it complements previously published results. The identification of different ion channels involved in Lhb neuronal excitability in naïve mice is interesting, yet there is no description of a potential functional alteration of these ion channels in susceptible mice.

1. It would be important to characterize, in naïve mice and in a few neurons, the expression of short- and long-RDP depending on the membrane potential of the neurons, as voltage-gated ion channels can also control the duration of the RDP.

[Our response to the comment]

Thank you for these very helpful suggestions. We added experiments for checking the hyperpolarizing prepulse dependency of the short- and long-RDP durations. RDPs were evoked by hyperpolarizing prepulse injections of -10 , -20 , -40 , and -100 pA in the presence of tetrodotoxin (0.5 μ M). The duration of long-RDPs progressively lengthened as the hyperpolarizing current was increased, reaching a plateau at approximately -40 pA (Fig. 5B). By contrast, the duration of short-RDPs was insensitive to the increase in hyperpolarizing current injections (Fig. 5A). These

responses to hyperpolarizing prepulses suggest that the generation of the long depolarizing phase is mediated by ion channels selectively expressed in long-RDP neurons but not in short-RDP neurons. The following pharmacological experiments supported this hypothesis, indicating that the long depolarization phase of long-RDPs was mediated by CNG channels.

This new data has been included in Figure 5A, B and is now described in the Results section (page 23, line 526–531).

2. Line 537 "These data suggest that at least a part of CNG channels is activated via the gap." To conclude this, they should prove that there is an occlusion of the LCD effect in the presence of Carb.

[Our response to the comment]

We examined whether *L-cis*-diltiazem would suppress long-RDPs further after the prior application of carbenoxolone. In a majority of LHb neurons (5/8), carbenoxolone largely suppressed the long depolarizing phase of the long-RDPs, converting the response to the short-RDP-like response. Subsequent *L-cis*-diltiazem application did not produce any additional suppression (Fig. 7E, G), indicating that most CNG channels are predominantly activated in the post-junctional cells. Meanwhile, in a smaller subset of neurons (3/8), carbenoxolone reduced more than half of the long depolarizing phase, with the remaining depolarization being suppressed by subsequent application of *L-cis*-diltiazem (Fig. 7F, G). This suggests that, in these neurons, CNG channels are present on both pre- and post-junctional cells. However, even in these cases, the contribution of pre-junctional CNG channels appeared to be minor compared to the post-junctional channels. Collectively, these data suggest that post-junctional CNG channels play a central role in generating the long depolarizing phase of long-RDPs in LHb neurons.

These new data have been included in Fig. 7E–G and are now described in the Results section (page 25, line 583–594).

3. Is there a reduction of T-type, SK, CNG current, or neuron-nonneuron connectivity in susceptible mice compared to resilient and naïve animals?

[Our response to the comment]

Our data indicate that short-RDPs are increased in susceptible mice (Fig. 3B). Notably, these short-RDPs in susceptible mice lacked the component mediated by CNG channels (as detailed in our response to your 4th comment) (Fig. 9A–C), suggesting an overall downregulation of CNG channel activity in susceptible mice.

To evaluate potential reductions in T-type VDCC and SK channel activity, we initially compared the amplitude of short-RDPs among the naïve, susceptible, and resilient mice, given that short-RDPs are not dependent on CNG channel activity (discussed in our response to your 4th comment) (Fig. 9G). We found that the short-RDP amplitude was comparable among these groups.

These results imply that the activity of T-type VDCCs and SK channels is comparable between the naïve, susceptible, and resilient mice, assuming that short-RDP generation in susceptible mice is mediated by these channels in a manner similar to normal mice (Fig. 5).

We further tested this assumption pharmacologically in susceptible mice. Similar to the RDPs in control mice, short-RDPs in susceptible mice were prolonged by apamin (Fig. 9H, I) and completely blocked by Z944 (Fig. 9H, J), confirming that short-RDPs are elicited by T-type VDCCs and shortened by SK channels. Taken together, these results reveal that the activities of T-type VDCCs and SK channels in generating short-RDPs are unchanged between control and susceptible mice. Instead, the increased prevalence of short-RDPs in susceptible mice is attributable to reduced CNG channel activity.

These new pharmacological data have been included in Figure 9A–C and 9G–J and are now described in the Results section (page 27, line 642– page 28, line 656).

4. To fully support the general conclusions of the paper, it would be important to describe the duration of RDP in susceptible and resilient mice using at least LCD and Carb/MFA blockers. If the general hypothesis is that CNG channels are reduced but not the other channels, we should expect an occlusion of the LCD effect on the duration of the RDP. If the generation of the long-RDP is due to neuron-nonneuron coupling, we could also predict an occlusion of the Carb/MFA effect.

[Our response to the comment]

As suggested by the reviewer, we applied LCD to short-RDPs in susceptible mice. We found that LCD did not affect the short-RDP durations in the susceptible mice (Fig. 9A, B), suggesting that CNG channels are not involved in the generation of the short-RDP in the susceptible mice. Furthermore, we performed a similar experiment using carbenoxolone and obtained the same result (Fig. 9A, C). Taken together with RNA sequencing data of *Cnga4* (Fig. 10A), the increased incidence of short-RDPs in the susceptible mice is attributable to reduced CNG channel activity. Whereas the increase in short-RDPs could also be explained by the downregulation of gap junctions, RNA sequencing data indicate that expression of major connexins remains unchanged by the social defeat stress (*Gjb1* [$p = 0.774$, FDR > 0.999], *Gjc2* [$p = 0.396$, FDR > 0.999], *Gjal* [$p = 0.847$, FDR > 0.999], *Gjb2* [$p = 0.328$, FDR > 0.999], *Gjb6* [$p = 0.619$, FDR > 0.999], *Gjc3* [$p = 0.0558$, FDR > 0.999], *Gjd2* [$p = 0.0464$, FDR = 0.988]). These data collectively suggest that the reduced CNG activity is a primary cause for increased incidence of short-RDPs.

We also conducted the same experiments on long-RDPs using resilient mice. In these mice, the effects of LCD and carbenoxolone on the long depolarizing phase of long-RDPs were comparable to those of control mice (Fig. 9D–F). These data suggest that the post-junctional CNG channels generate the long depolarizing phase of long-RDPs even in resilient mice.

These new data have been included in Figure 9A–F and are now described in the Results section (page 27, line 642– page 28, line 656).

Minor Points:

1. The title is not completely appropriate to the findings provided in this study. Indeed, they did not investigate neuron-nonneuron electrical coupling in mice submitted to chronic stress. Yet, if they address the major points, it will be appropriate.

[Our response to the comment]

As suggested by the reviewer, we have incorporated new data to address the major points raised. With these additions, we believe the current title accurately reflects the scope and findings of the present study.

2. Figure 1L is not discussed in the results.

[Our response to the comment]

We have now moved Fig. 1L to Fig. 3D–G and added text referring to Fig. 3D–G to the Results section (page 21, line 429–496).

3. The term "frequency adaptation index" is misleading, as we would expect to have high adaptation properties when close to 1. It would be great to define it better in the results paragraph.

[Our response to the comment]

We have now calculated the frequency adaptation index as follows:

$$\text{Frequency adaptation index} = \frac{|(\text{Initial firing frequency}) - (\text{Final firing frequency})|}{\text{Initial firing frequency}}$$

Where ‘initial firing frequency’ is the inverse of the first and second spike intervals, and ‘final firing frequency’ is the average firing frequency during the 200 ms immediately before the offset of depolarization. We think this index is more adequate for the present analysis because this will increase to 1 with the severity of frequency adaptation.

This is now explained in the Methods section (page 12, lines 267–274).

4. The description of the four neuron types (HF-short-RDP, HF-long-RDP, LF-short-RDP, LF-long-RDP) is interesting because it takes into account two main electrophysiological properties of LHb neurons. Unfortunately, in Figure 3, the quantification is based only on the duration of RDP or the frequency of the neurons, which raises questions about the pertinence of the cluster analysis. It would be interesting to add the quantification of the neurons in the four groups when comparing

naïve, susceptible, and resilient mice.

[Our response to the comment]

We have now added a bar graph quantifying the four neuron groups as Fig. 3A. Although overall distributions were not statistically significant using the χ^2 test, residuals analysis revealed a significant decrease in the incidence of the long-RDP&LF group in the susceptible mice.

These new data are described in the Results section (page 21, lines 479–483).

5. Line 532 "Because previous studies have demonstrated that CNG channel activation is affected by Ni^{2+} ." Considering this, the authors should use a selective blocker of T-type channels (Z944) to really conclude the role of T-type in the generation of bursts.

[Our response to the comment]

We examined the effects of Z944 on RDPs. Z944 effectively abolished both short- and long-RDPs (Fig. 5H, 9H–J), which were identical to the results using high concentration of Ni^{2+} (Fig. 5C–G). These data strongly confirm the contribution of T-type VDCC to generate RDPs.

These new data have been included in Fig. 5H, 9H–J and are described in the Results section (page 23, line 541–542 and page 28, line 651–654).

References in this letter

- Broillet MC & Firestein S. (1997). Beta subunits of the olfactory cyclic nucleotide-gated channel form a nitric oxide activated Ca²⁺ channel. *Neuron* **18**, 951–958.
- Chang SY & Kim U. (2004). Ionic mechanism of long-lasting discharges of action potentials triggered by membrane hyperpolarization in the medial lateral habenula. *J Neurosci* **24**, 2172–2181.
- Ito H, Nozaki K, Sakimura K, Abe M, Yamawaki S & Aizawa H. (2021). Activation of proprotein convertase in the mouse habenula causes depressive-like behaviors through remodeling of extracellular matrix. *Neuropsychopharmacology* **46**, 442–454.
- Matulef K & Zagotta WN. (2003). Cyclic nucleotide-gated ion channels. *Annu Rev Cell Dev Biol* **19**, 23–44.
- Meighan PC, Meighan SE, Rich ED, Brown RL & Varnum MD. (2012). Matrix metalloproteinase-9 and -2 enhance the ligand sensitivity of photoreceptor cyclic nucleotide-gated channels. *Channels (Austin)* **6**, 181–196.
- Meighan SE, Meighan PC, Rich ED, Brown RL & Varnum MD. (2013). Cyclic nucleotide-gated channel subunit glycosylation regulates matrix metalloproteinase-dependent changes in channel gating. *Biochemistry* **52**, 8352–8362.
- Oh SJ, Park JH, Han S, Lee JK, Roh EJ & Lee CJ. (2008). Development of selective blockers for Ca²⁺(+)-activated Cl⁻ channel using *Xenopus laevis* oocytes with an improved drug screening strategy. *Mol Brain* **1**, 14.
- White MM & Aylwin M. (1990). Niflumic and flufenamic acids are potent reversible blockers of Ca²⁺(+)-activated Cl⁻ channels in *Xenopus* oocytes. *Mol Pharmacol* **37**, 720–724.

Dear Dr Hashimoto,

Re: JP-RP-2025-287286R1 **"Neuron-non-neuron electrical coupling networks are involved in chronic stress-induced electrophysiological changes in lateral habenular neurons"** by Kenji Yamaoka, Kanako Nozaki, Meina Zhu, Haruhi Terai, Kenta Kobayashi, Hikaru Ito, Miho Matsumata, Hidenori Takemoto, Shinya Ikeda, Yusuke Sotomaru, Tetsuji Mori, Hidenori Aizawa, and Kouichi Hashimoto

We are pleased to tell you that your paper has been accepted for publication in The Journal of Physiology.

Yours sincerely,

Katalin Toth
Senior Editor
The Journal of Physiology

If you would like to receive our 'Research Roundup', a monthly newsletter highlighting the cutting-edge research published in The Physiological Society's family of journals (The Journal of Physiology, Experimental Physiology, Physiological Reports, The Journal of Nutritional Physiology and The Journal of Precision Medicine: Health and Disease), please click this link, fill in your name and email address and select 'Research Roundup':
<https://www.physoc.org/journals-and-media/membernews>

- You can help your research get the attention it deserves! Check out Wiley's free Promotion Guide for best-practice recommendations for promoting your work at: www.wileyauthors.com/eoo/guide. You can learn more about Wiley Editing Services which offers professional video, design, and writing services to create shareable video abstracts, infographics, conference posters, lay summaries, and research news stories for your research at: www.wileyauthors.com/eoo/promotion.

EDITOR COMMENTS

Reviewing Editor:

The additional data and information provided in the revision have significantly improved the manuscript.

REFEREE COMMENTS

Referee #1:

The authors performed the additional experiments and added more information and discussion than expected to the revised MS and the reviewer's concerns are fully addressed.

The abstract diagram is impressive to catch the essence of the findings.

The The revised MS is acceptable and surely suitable for publication in JP.

Referee #2:

In this study, Yamaoka et al. describe a neuron-non-neuron electrical coupling network that contributes to chronic stress-induced electrophysiological changes in lateral habenular (LHb) neurons. Their findings reveal a novel adaptive mechanism in LHb electrophysiology, specifically involving coupling with oligodendrocytes.

By examining mice subjected to social defeat stress, the authors demonstrate that LHb neuronal activity is modulated under stress. Using pharmacological and genetic approaches, they identify distinct ion channels involved in regulating short and long rebound depolarization (RDP), including T-type, SK, and CNG channels, as well as gap junctions. These channels likely contribute to neuron-non-neuron coupling networks that regulate RDP in LHb neurons.

Beyond characterizing RDP in naïve mice, the study also shows that stress susceptibility alters the proportion of short- and long-RDP neurons, supporting the broader hypothesis of LHb hyperactivity in stressed mice. These findings challenge the classical model of LHb burst activity by uncovering the role of specific ion channels in modulating burst firing, providing new insights into the physiological mechanisms underlying stress-induced LHb dysfunction.

In this study, Yamaoka et al. describe a neuron–non-neuron electrical coupling network that contributes to chronic stress-induced electrophysiological changes in lateral habenular (LHb) neurons. Their findings reveal a novel adaptive mechanism in LHb electrophysiology, specifically involving coupling with oligodendrocytes.

By examining mice subjected to social defeat stress, the authors demonstrate that LHb neuronal activity is modulated under stress. Using pharmacological and genetic approaches, they identify distinct ion channels involved in regulating short and long rebound depolarization (RDP), including T-type, SK, and CNG channels, as well as gap junctions. These channels likely contribute to neuron–non-neuron coupling networks that regulate RDP in LHb neurons.

Beyond characterizing RDP in naïve mice, the study also shows that stress susceptibility alters the proportion of short- and long-RDP neurons, supporting the broader hypothesis of LHb hyperactivity in stressed mice. These findings challenge the classical model of LHb burst activity by uncovering the role of specific ion channels in modulating burst firing, providing new insights into the physiological mechanisms underlying stress-induced LHb dysfunction.